# Prematurely terminated intron-retaining mRNAs invade axons in SFPQ null-driven neurodegeneration and are a hallmark of ALS

Richard Taylor [1] ✉, Fursham Hamid [1], Triona Fielding[1], Patricia M. Gordon[1], Megan Maloney[1], Eugene V. Makeyev [1] & Corinne Houart [1] ✉

Loss of SFPQ is a hallmark of motor degeneration in ALS and prevents maturation of motor neurons when occurring during embryogenesis. Here, we show that in zebrafish, developing motor neurons lacking SFPQ exhibit axon extension, branching and synaptogenesis defects, prior to degeneration. Subcellular transcriptomics reveals that loss of SFPQ in neurons produces a complex set of aberrant intron-retaining (IR) transcripts coding for neuron-specific proteins that accumulate in neurites. Some of these local IR mRNAs are prematurely terminated within the retained intron (PreT-IR). PreT-IR mRNAs undergo intronic polyadenylation, nuclear export, and localise to neurites in vitro and in vivo. We find these IR and PreT-IR mRNAs enriched in RNAseq datasets of tissue from patients with familial and sporadic ALS. This shared signature, between SFPQ-depleted neurons and ALS, functionally implicates SFPQ with the disease and suggests that neurite-centred perturbation of alternatively spliced isoforms drives the neurodegenerative process.

Regulation of local transcriptomic landscapes in axons and dendrites is crucial for their development and survival[1–3]. The neuronal-enriched Splicing Factor, SFPQ (aka PSF), is essential for motor circuit development and its misregulation is strongly linked to neurodegeneration[4–8]. While SFPQ is predominantly nuclear, involved in several RNA regulatory processes, the protein also localises to axons and dendrites. Such non-nuclear pools control axonal growth and synaptogenesis[4], and facilitate axonal and dendritic localisation of RNA granules by associating with a kinesin motor protein[9–11].

Introns are typically removed from pre-mRNAs during splicing. Transcripts in which one or multiple introns are retained, are classically associated with degradation or nuclear detention[12]. Recently, retained introns have been observed outside of the nucleus, implicated in promoting mRNA transport[13]. IR transcripts can also be translated, producing functionally relevant small amounts of truncated protein isoforms, using an in-frame stop codon inside the retained intron[14–17].

Studies analysing patient tissues and cellular and animal models of amyotrophic lateral sclerosis (ALS) and frontotemporal lobar dementia (FTLD) have found nuclear depletion and/or cytoplasmic accumulation/aggregation of SFPQ[4,18–20]. Depletion of SFPQ protein has been shown to cause mis-splicing, accompanied by tauopathy and an FTLD-like phenotype[6,7,21]. Missense mutations within *sfpq* found in a small number of ALS patients have also been shown to cause neurodegeneration ALS[4,18]. Studies across sporadic and familial cases have established SFPQ misregulation as a hallmark of ALS/FTLD[5]. However, despite the severe impact that SFPQ depletion has on motor neurons in development and on neuronal survival in adult, very little is known about the molecular events triggered by its loss.

In this study, we provide such molecular understanding. We establish for the first time a functional relationship between loss of

[1]Centre for Developmental Neurobiology and Medical Research Council Centre for Neurodevelopmental Disorders, Institute of Psychiatry, Psychology & Neuroscience, Guy's Campus, King's College London, London SE1 1UL, UK. ✉e-mail: richard.taylor@kcl.ac.uk; corinne.houart@kcl.ac.uk

SFPQ and the emergence of intron-retaining mRNAs enriched in ALS. We also reveal that this increase specifically impacts axons and dendrites. Strikingly, a great proportion of these aberrant IR mRNAs are prematurely terminated within the retained intron (PreT-IR) and yet are stable and transported in axons and dendrites. Unexpectedly, prematurely terminated mRNAs are not systematically retained in the nucleus and degraded. Neurite-specific IR and PreT-IR transcripts disrupt the local balance of alternatively spliced mRNA isoforms, likely impacting the local proteome and affecting axonal and dendritic homeostasis and regulation of synaptic activity.

We find these IR and PreT-IR mRNAs enriched in familial and sporadic ALS patient brain samples and iPSC-derived neurons, establishing causality from SFPQ loss-of-function to IR dysregulation in ALS/FLTD, and identifying PreT-IR as a novel hallmark of the disease. Moreover, the presence of the same abnormal mRNAs in patients and in our SFPQ loss-of-function animal model indicates that these are likely localised in neurites in patient neurons, thereby implying neurite-centred neurodegeneration by local misexpression of SFPQ-regulated splice isoforms.

## Results

### SFPQ-depleted developing neurons exhibit axonal defects and downregulation of axonal transcripts

Zebrafish and mouse embryos lacking SFPQ in all tissues exhibit profound defects in nervous system development, including failed motor axon extension[4,22]. We previously showed that zebrafish *sfpq* null (−/−) motor neurons transplanted into a wild-type embryo fail to form an axon at the normal time (by 24 hours post-fertilisation, hpf)[4]. To assess whether this inability is a mere delay or a profound change in neuronal development, we followed by in vivo time-lapse imaging, the fate of transplanted null motor neurons beyond 24 hpf, up to free-swimming and feeding larvae (120 hpf). Over this extended time course, we find that in over 30% of cases, null motor neurons do extend axons, with substantial delay compared to sibling (control; *sfpq* +/+ and +/−) counterparts (Fig. 1a, b and Supplementary Fig. 1a–d). Null motor axons form very few, if any, branches (Fig. 1c) and far fewer synapses (Fig. 1d and Supplementary Fig. 1e–f). The remaining null motor neurons did not extend axons. Null motor neurons display classical signs of degeneration from 72 hpf (Supplementary Fig. 1g–i). Axons show fragmentation, a typical 'dying back' indicator of neuronal death reminiscent of that seen in neurodegenerative disorders, including in ALS (Supplementary Fig. 1h)[23–26]. Motor neuron somas never seen with axons also eventually die (Supplementary Fig. 1i).

Given the profound axonal defects we observed in mutant neurons, we sought to identify the changes in local transcriptome driving these abnormalities. Our previous work showed that the introduction of a cytoplasmic SFPQ variant into null zebrafish embryos rescues motor axon extension, motility defects, and expression of transcripts encoding axonal proteins unambiguously demonstrating the importance of non-nuclear roles[4]. To enable isolation of neurites in large quantities we established a zebrafish primary culture method allowing long-term neuronal cultures derived from dissociated 24 hpf sibling and null embryos. Over six days in vitro (DIV), dissociated cells become increasingly organised into clusters, and extend neurites to other cells within the cluster and neighbouring clusters (Supplementary Fig. 2a, b). As expected, based on our in vivo dataset, null motor neuron projections showed significantly fewer branches at all DIV stages (Supplementary Figs. 2h, 3d). Analyses revealed that sibling and null cultures were most similar at DIV2 (Supplementary Figs. 2c–f, 3a–c).

To isolate neurites from sibling and null neurons for transcriptomic analysis, we performed primary culture in transwell inserts until DIV2 (Fig. 1e and Supplementary Fig. 4a). Cellular and neurite contents were separately collected from sibling and null inserts for RNAseq profiling. Principal component analyses visualising expression counts of enriched genes indicated that samples were consistently

segregated according to genetic background and compartment. Pronounced compartment-specific transcriptomes were observed, with subtle differences between sibling and null samples (Fig. 1f, g; Supplementary Figs. 4c–d, f and Supplementary data 1).

Comparing sibling and null neurite expression profiles revealed that most changes were downregulation (73%) (Supplementary Fig. 4e and Supplementary data 2, 3). 597 annotated genes were downregulated in null neurites (Supplementary data 2), enriched for GO terms such as axon guidance, synaptic transmission, and neuron development (Supplementary data 9). Given that the cellular compartment contains both cell bodies and neurites (Supplementary Fig. 4a), we would expect to detect this downregulation in both compartments, and indeed all these genes also show downregulation in the null cellular compartment. However, 185 genes (31%) showed substantially stronger downregulation in the neurite compartment, than in the cellular compartment (Log2FC neurite - Log2FC cellular = <−0.2; Supplementary data 2, Supplementary Fig. 5a), enriched for GO terms associated with axonogenesis (Supplementary data 9). RT-qPCR analysis of three neurite-specific downregulated mRNAs that encode neuronal proteins (*nova2*, *smap1* and *gnao1a*) validated our RNAseq findings, with downregulation in both compartments, more pronounced in null neurites (Fig. 1h–j). Further validation of transcript downregulation in neurites in primary culture was performed using RNAscope targeting *nova2*, encoding a neuronal alternative splicing factor important for axon guidance. Quantification of high-resolution imaging revealed that *nova2* puncta in neurites are significantly reduced in number in null culture (Supplementary Fig. 5b–e). Transcript decreases specifically in neurites indicates a role for SFPQ in controlling the local transcriptome landscape in neurites via transport and/or stabilisation of select mRNAs.

### SFPQ-depletion results in neurite-specific localisation of transcripts retaining long SFPQ-binding introns

Given that SFPQ is a splicing factor, we next analysed mRNA splice variants in sibling and null compartmentalised samples. To do this, we ran our RNAseq dataset through VAST-TOOLS, a pipeline that detects and quantifies various types of alternative splicing and analyses their differences between samples[27]. Many splicing events were detected, with considerable compartment-specific bias (Supplementary data 4). Cassette exon usage was expectedly the most abundant type of splicing change in absence of SFPQ when comparing cellular compartments (Supplementary Fig. 6a top, b, c). Contrastingly, comparing neurite compartments revealed intron retention as the dominant form of splicing change, being mostly increases in IR transcripts (Supplementary Fig. 6a bottom, d, e). This observation demonstrates further the importance of SFPQ in controlling the local transcriptome in developing neurites, preventing local presence of IR mRNAs.

We deepened our search and analysed our dataset using IRFinder, a program specifically exploring intron retention[28]. In vast-tools, 195 increased (in 189 genes) and 35 decreased (in 35 genes) intron retention events were observed in neurites of null neurons (cut-off: ≥10% changes in the IR transcripts, MV.[dPsi]_at_0.95 = ≥0.01, Supplementary data 5). In IRFinder, remarkably similar proportions of increased and decreased intron retention events were observed (Fig. 2a). 446 introns were retained more abundantly in null neurites, while 86 introns were less retained (cut-off: IR ratio changing by >0.1 in either direction, $p < 0.05$; Supplementary data 5). The majority of increases in intron retention were neurite-specific (Fig. 2b and Supplementary Fig. 7a). The extent to which these introns are retained in transcripts of their respective genes varies substantially, as does the extent of increase in retention, with a median increase in IR ratio of 0.17 (Fig. 2c, d). Some transcripts also showed intron retention in sibling samples, albeit to a much lesser extent, suggesting a possible function for these introns in normal neurites (Supplementary data 5). GO term biological processes for the genes associated with the retained introns included

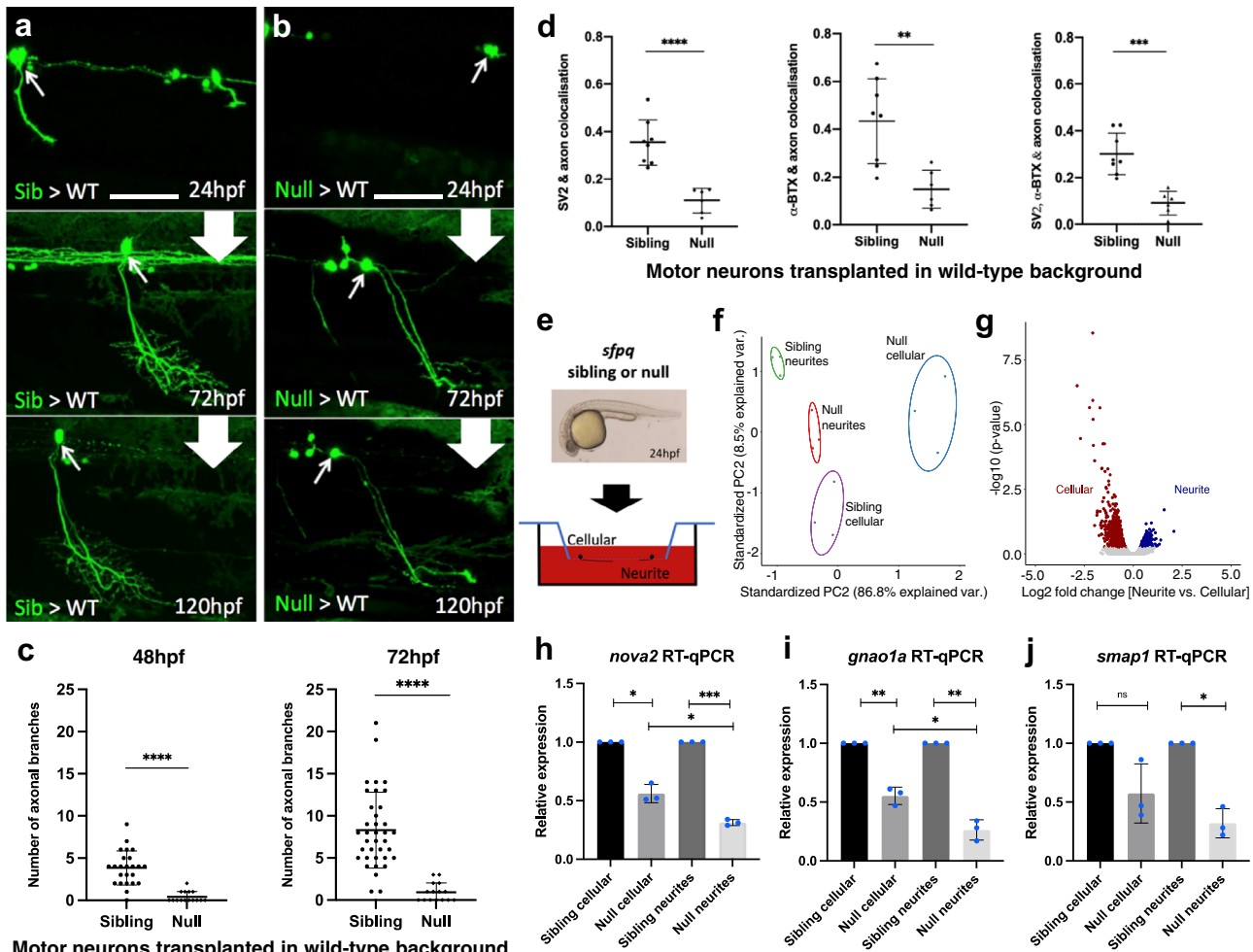

**Fig. 1 | Characterisation of the impact of SFPQ-depletion on neurites.**
**a**, **b** Confocal z-projections (150 μm), lateral view, anterior to the left, of transplanted *sfpq*; tg(mnx1:GFP) sibling (**a**) & null (**b**) motor neurons in wild-type hosts. Scale bar 25 μm. **c** Quantification of all branches on transplanted sibling and null GFP + motor neurons at 48 hpf (Left) and 72 hpf (Right). Each datapoint represents an individual neuron. Two-tailed, unpaired t tests with Welch's correction. Means plotted with SD. (48 hpf) Sibling: 24 neurons, 9 hosts (6 donors), Null: 15 neurons, 4 hosts (4 donors), ****$p$ < 0.0001. (72 hpf) Sibling: 37 neurons, 14 hosts (11 donors), Null: 15 neurons, 4 hosts (4 donors), ****$p$ < 0.0001. **d** Quantification of axon colocalisation with synaptic markers. The portion of axon overlapping with each marker was calculated. Each point represents an axon of a different neuron. Sibling, n: 8 neurons from 3 hosts (2 donors); null n: 6 neurons from 2 hosts (2 donors). Two-tailed Unpaired ttests performed with Welch's correction. Mean plotted with SD. (Left) ****$p$ < 0.0001, (Middle) **$p$:0.0024, (Right) ***$p$:0.0001. **e** Sibling versus null

cells from dissociated *sfpq*, tg(mnx1:GFP) embryos were plated onto the upper membrane surface transwell inserts. Neurites can extend through 1 μm pores and adhere to the lower surface. **f** Principal component analysis plot of the 12 cellular and neurite RNA samples using 1359 highly variant genes. **g** Plot showing clusters for transcripts enriched in sibling cellular and neurite compartments. Cellular: FC > 1.5, $p$ < 0.05; 1030 genes. Neurite: FC > 1.25, $p$ < 0.05; 329 genes ($p$-values calculated using two-sided Wald tests). **h**–**j** RT-qPCR validation of neurite-specific gene downregulation in null ($n$ = 3 biological replicates, blue dots). Expression normalised to relevant compartment *actl6a* expression. Sibling expressions were set to 1. Two-tailed unpaired ttests with Welch's correction. Means plotted with SD. **h** *nova2*, cellular, *$p$:0.0103; neurite, ***$p$:0.0004; cellular vs neurite, *$p$: 0.0232. **i** *gnao1a*, cellular, **$p$:0.009; neurite, **$p$:0.0045; cellular vs neurite, *$p$:0.0121. **j** *smap1*, cellular, $p$ = 0.0989; neurite, *$p$:0.0111. Source data are provided as a Source Data file.

nervous system development, axon guidance and neuron projection development (Supplementary data 9). Local translation in neurites is reported as the predominant source of protein for many of the affected transcripts[1] (Supplementary data 6). No GO terms were enriched for the few transcripts showing decreased intron retention in null neurites. Both analyses revealed significant differences in intron length between retained and non-retained introns, with very long introns being overrepresented and medium length introns underrepresented in the retained group (Fig. 2e, g and Supplementary Fig. 7b). In line with this observation, retained introns are more frequently positioned relatively 5′ within genes (Fig. 2f, Supplementary Fig. 7c)[29,30]. We validated the intron retention phenotype by RT-qPCR (Supplementary Fig. 7e–g).

SFPQ has been shown to bind introns, particularly those that are very long[22]. We therefore utilised an *Sfpq* (cross-linking

immunoprecipitation) CLIP dataset generated from embryonic mouse cortex[22], to assess whether mouse homologues of the aberrantly retained introns are preferential SFPQ binding targets. Our analysis revealed substantially more SFPQ CLIP peaks associating to the retained introns compared to controls (Fig. 2h, Supplementary fig. 7d). This indicates that retained introns increased in null neurites are binding targets of SFPQ, and that the regulation of these introns is conserved across species.

## Neurite localised IR mRNAs are frequently prematurely terminated and polyadenylated within the intron

A known consequence of *sfpq* inactivation in the nucleus is long gene transcriptopathy, where elongating Pol-II drop-off results in premature transcriptional termination, particularly prominent amongst long intron-containing genes[22]. Consistent with this, we find a correlation

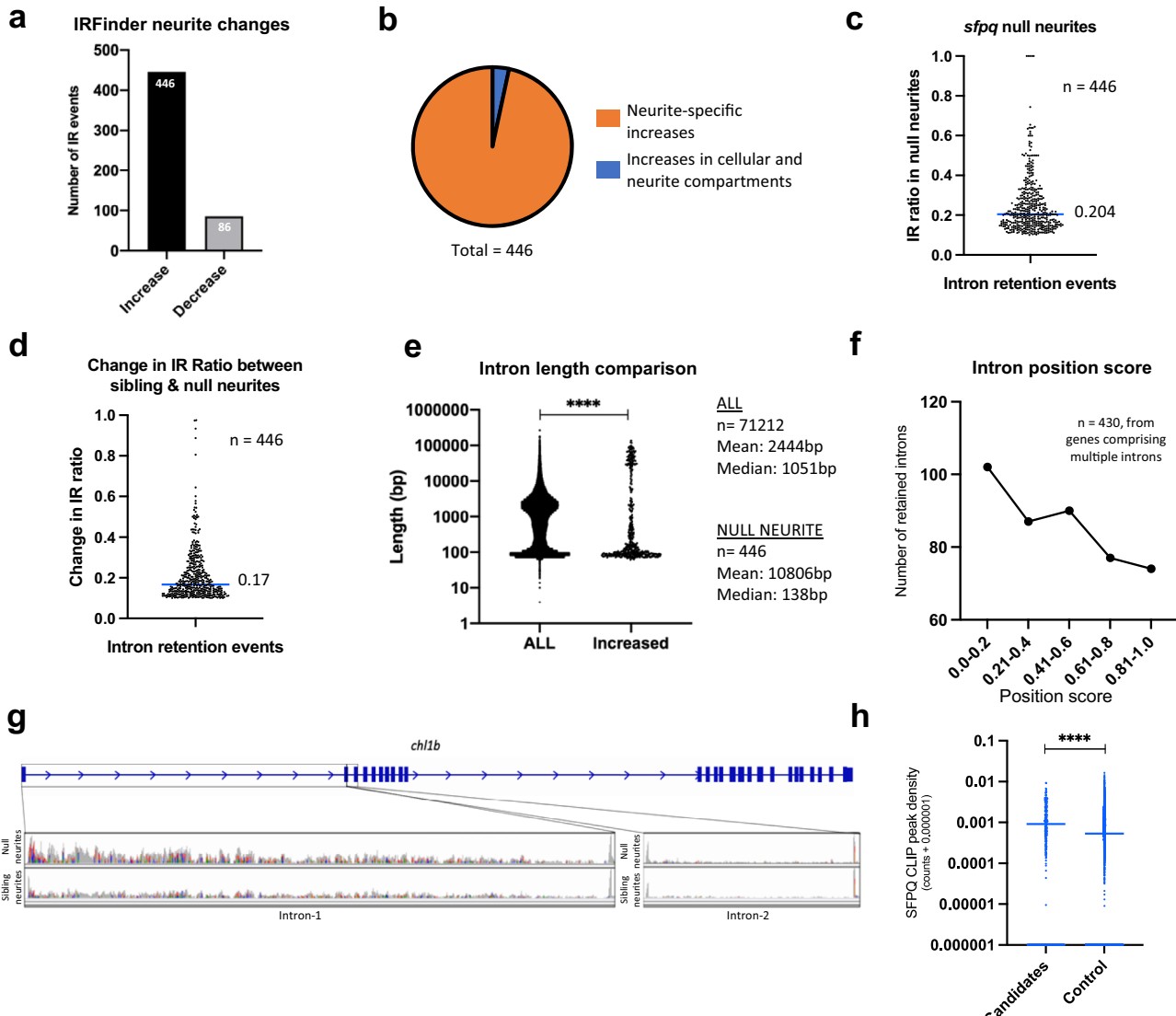

**Fig. 2 | Loss of SFPQ modifies distribution of neurite-specific retained introns.** **a** Number of intron retention increases and decreases observed in null neurites identified in IR Finder (>0.1 change in IR Ratio, Audic & Claverie test, *p* < 0.05) (**b**) Proportion of intron retention increases that are neurite-specific (change in neurite IR Ratio is >0.05 above that in the cellular compartment). 15 intron retention events (blue slice) increase similarly in both the neurite and cellular compartments. **c** Proportion of transcripts for each of the 446 genes in (**a**) that are intron-retaining in null neurites. Median IR ratio = 0.204 (i.e. ~20% of all transcripts of a given gene). **d** Change in IR ratio from sibling neurites to null neurites (median increase = 0.17).

**e** Length of all introns from the annotated genome compared with the length of the 446 introns in (**a**). Two-sided wilcoxon-signed rank test, ****p* < 0.0001. **f** Positional information of the retained introns in (**b**). **g** Cell adhesion like molecule L1-like (**b**) (*chl1b*) RNAseq BAM files showing increased intron-1(102 kbp long) retention in null neurites, and no retention of intron-2 (used as control) in either sibling or null neurites. **h** CLIP peak density scores for 356 mouse homologues of zebrafish introns identified by IRFinder and vast-tools as being more retained in null neurites, and of 7203 control non-retained introns from the same genes. Two-tailed Mann–Whitney test, ****p* < 0.0001.

between retained intron length and gene downregulation in null cellular and neurite compartments (Supplementary Fig. 7h). These long, downregulated genes were enriched for cell projection morphogenesis, neuron projection morphogenesis, neuron development and related GO terms (Supplementary data 9). We perfomed a systematic analysis of the IR transcripts' RNAseq reads, binning exonic sequence reads according to their 5' to 3' position within the transcript. The reads mapping to each bin were then quantified, uncovering two types of IR transcript robustly enriched in null neurites: 'classic' full-length IR mRNAs (no slope, 31% of neurite-specific IR genes - FDR < 0.05), and truncated 'sloping' IR mRNAs (69% of neurite-specific IR genes - FDR < 0.05; Fig. 3a, b, c). For the latter group of truncated 'sloping' IR mRNAs, more reads were present for 5' exons compared to 3'exons, producing a negative slope gradient value, with a strong correlation between intron length and slope gradient (Fig. 3c, d; Supplementary

Fig. 8 and Supplementary data 7). Retained introns in the sloping IR transcripts are, on average, longer than those in classic IR mRNAs (Fig. 3e) and tend to sit closer to the 5' end of their host transcripts (Fig. 3f).

Together, these data indicate that large numbers of prematurely terminated intron-retaining (PreT-IR) transcripts localise to null neurites, a phenomenon highly unexpected as premature transcriptional termination is assumed to lead to mRNA nuclear sequestration and/or degradation. To investigate whether these transcripts are stabilised by alternative polyadenylation (APA), we searched our 24 hpf sibling and null 3'-mRNAseq datasets[29]. We found in null, an enrichment of proximal poly(A) 3'mRNAseq read clusters preceded by polyadenylation consensus sequences (AATAAA or other reported variations) within the retained introns of 'sloping'/PreT-IR transcripts (Supplementary Fig. 9a, b middle & right, and Fig. 4a), revealing frequent use of such

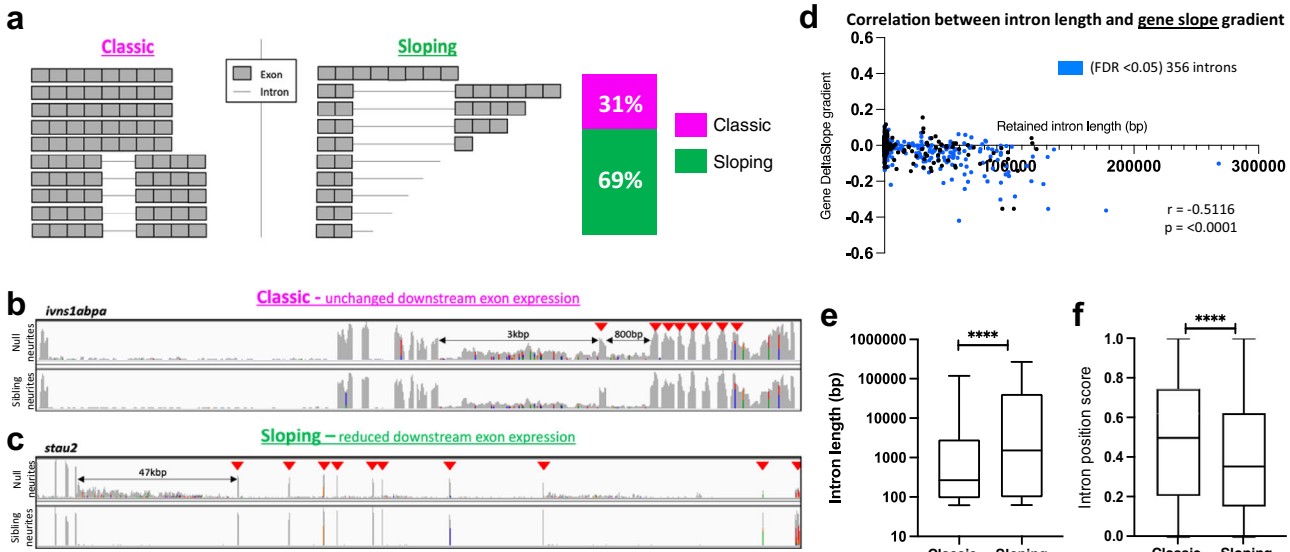

**Fig. 3 | Two types of intron-retaining transcripts become neurite localised in null neurons. a** Schematic representation of the two types of IR-transcripts enriched in null neurites. The stacked bar graph shows the proportion of genes expressing 'classic' full-length IR transcripts (magenta) and 'sloping' prematurely terminated IR transcripts (green) transcripts; *n* = 356 genes, FDR < 0.05.
**b**, **c** Example gene plots for each of the two types of IR-transcript in null neurites. Arrows indicate exons downstream of the retained intron/s. **b** *ivns1abpa*, intron-8-retaining, and intron-9-retaining transcripts with no change in downstream exons expression. **c** *stau2*, intron-5-retaining transcripts showing reduced expression of downstream exons. **d** Correlation between retained intron length and 5′→3′ exonic read slope gradient. Gradients calculated for the 356 genes encoding IR-transcripts

(blue datapoints, FDR < 0.05). Two-sided Pearson correlation coefficient, r: −5116, *p* < 0.0001. **e** Box and whiskers plot showing the difference in length of retained introns in each IR-transcript category. The box boundaries represent the upper and lower quartiles, and the middle line is the median. Whiskers show maximum and minimum values. Two-sided Wilcoxon-signed rank test, ****\**p* < 0.0001. *n* = 570 introns from multiple intron-containing genes. **f** Box and whiskers plot showing the difference in position of the retained introns in transcripts for each IR-transcript category. The box boundaries represent the upper and lower quartiles, and the middle line is the median. The whiskers show the maximum and minimum values. Two-sided Wilcoxon-signed rank test, ****\**p* < 0.0001. *n* = 570 introns from multiple intron-containing genes.

intronic polyadenylation sequences (iPAS). Intronic polyadenylation (IpA) is a relatively uncommon subclass of ApA frequently occurring in immune cells[31] and associated with cancer[32]. As well as an increased number of cleavage/polyadenylation-read clusters, increases in site usage were observed in absence of SFPQ (Fig. 4b middle & right). Such effects were not observed for 'classic' full-length IR mRNAs (Fig. 4b left, & Supplementary Fig. 9a, b left). The presence of transcripts truncated within their retained introns and stabilised was validated by 3′RACE for *igdcc3* (Fig. 4c, d) and *stau2* (Supplementary Fig. 9d, e). These PreT-IR transcripts encode truncated proteins with in-frame termination codons positioned early in the intron.

We next sought to verify the presence of PreT-IR transcripts and their neurite localisation in *sfpq* embryos. We performed HCR, a form of in situ hybridisation permitting visualisation of RNAs at subcellular resolution, on 24hpf *sfpq* sibling versus null embryos. Two probe sets were designed targeting intron-2 of *igdcc3*, the first against the proximal-most 10,000-nucleotides of the intron (5′) and the second against the distal-most 10,000-nucleotides (3′). Both probe sets showed strong forebrain expression in accordance with *igdcc3* expression reported on ZFIN for embryos of this stage (Supplementary Fig. 9g). The number of 5′ and 3′ intron-2 puncta in axons (Fig. 4e, f) was quantified. Both 5′ and 3′ intron labelling increase in null axons indicating enrichment of 'classic' intron-2-retaining *igdcc3* transcripts (Fig. 4g). Furthermore, much larger increases of the 5′ intron probe were observed compared to the 3′ probe, indicating presence of PreT-IR transcripts (Fig. 4h).

In summary, loss of SFPQ in neurons leads to neurite localisation of PreT-IR transcripts through nuclear Pol-II drop off, intronic polyadenylation, nuclear export and, either abundant transport to neurites or selective protection from degradation in neurites. These likely generate truncated proteins locally that interfere with normal functions.

## Neurite IR and PreT-IR transcripts are enriched in ALS samples

An emerging hallmark of ALS/FTLD, increased retention of specific introns in spliced mRNAs, has been observed across a range of genetic backgrounds, in cellular and animal models and patient tissue. Key affected introns are common across multiple datasets[18,19,33,34]. A functional connection between SFPQ misregulation and aberrant intron retention in ALS/FTLD has not yet been explored. To explore whether aberrant intron retention may be caused by SFPQ misregulation observed in patients, we assessed whether common ALS-associated introns were found in zebrafish null samples. We collated a list of 201 candidate human and mouse introns persistently changing their retention status across ALS samples[19,33,35–38] (Supplementary data 8). Like our retained introns in zebrafish neurites, we found that mouse homologues of ALS-associated introns bind SFPQ significantly more than control non-retained introns from the same genes[22] (Fig. 5a).

We were able to identify 134 zebrafish homologues from the 201 ALS-associated introns. We found that *sfpq*-intron-9 retention (the most reported in ALS) was significantly increased in null samples, and more so in neurites (IRFinder: p.diffBH $1.74 \times 10^{-9}$, Fig. 5b, c, Supplementary Fig. 10a). Although dramatically downregulated in our *sfpq* mutant[4], a detectable pool of intron-9-retaining *sfpq* mRNA is present. Increased intron-9-retention in the null and strong SFPQ binding to intron-9 (Fig. 5b) is suggestive of self-regulation. We found 25 other ALS-associated introns showing retention in our dataset, changing in null but missing statistical cut-offs. For these, we identified increases by RT-qPCR, again most pronounced in neurites (Fig. 5d, e; Supplementary Fig. 10b and Supplementary Table 1). The overlap is magnitudes above that expected for two unrelated data samples, truly remarkable given that we compared early neuronal development in fish with adult human and mouse postnatal tissue and differentiated neurons.

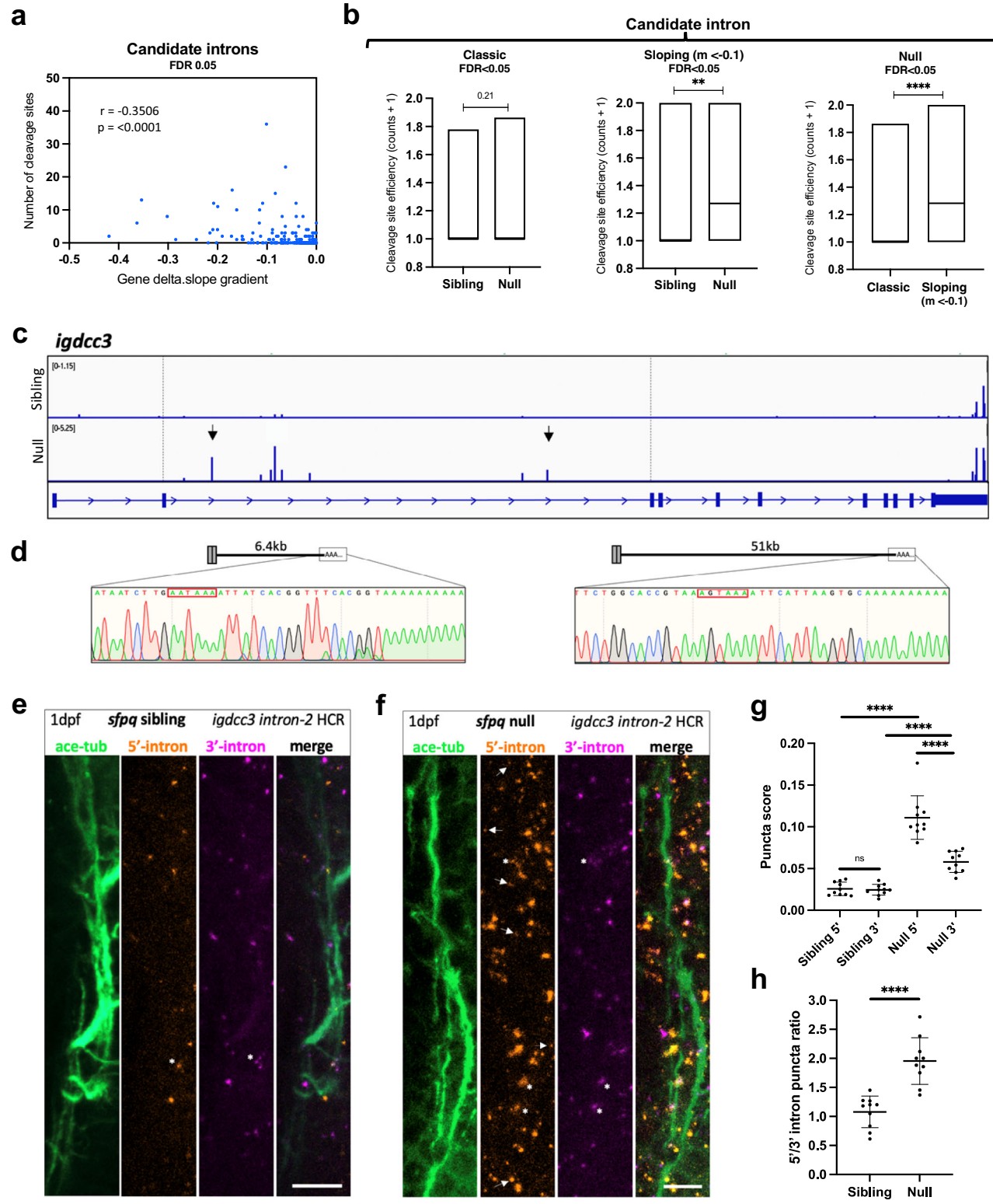

While intron retention and SFPQ misregulation are hallmarks of ALS/FTLD, to date, transcriptopathy and intronic polyadenylation have not been associated with neurodegenerative disease. To explore whether the PreT-IR phenotype uncovered in zebrafish neurites is also a hallmark of ALS, we analysed six total RNAseq datasets from ALS patients, mouse tissues and cellular models[33,36–40]. These datasets show variety in the genetic basis of ALS, representing sporadic and *FUS* mutation cases. Searching among genes >100 kb in length, we identified 64 genes exhibiting significant transcriptopathy compared to their

control counterparts. Of these, 21 genes were overlapping in 2 or more of the ALS datasets, with *EXT1* showing significant sloping in 4 datasets, and *TOX, FAM110B, SSBP2* and *CRHR1* each sloping in 3 datasets (Supplementary Table 2). Mouse homologues of 20 out of 21 of the genes were SFPQ binding, with most (17/21) exhibiting very strong binding (Supplementary Fig. 11). Impressively, homologues for 13 of the 21 genes show neurite-enriched PreT-IR in zebrafish null neurons. The other 8 just miss the statistical cut-off. Transcripts truncated within long retained introns, stabilised, and localised to null neurites

**Fig. 4 | PreT-IR transcripts undergo cleavage/polyadenylation and neurite localisation in SFPQ-deprived neurons. a** Correlation between RNAseq read slope gradient in null and number of cleavage sites in introns of sloping IR-transcripts. Two-tailed Pearson correlation coefficient, r: −0.3506, $p < 0.0001$ ($n = 247$ sloping IR-transcripts, FDR < 0.05). **b** Efficiency of cleavage site usage in classic (Left; ($n = 107$, FDR < 0.05)) and sloping (Middle; ($n = 37$, $m < −0.1$, FDR < 0.05)) IR-transcripts in sibling and null samples. (Right) Efficiency of cleavage site usage in retained introns between classic versus sloping IR-transcripts in null samples. Floating bar plots where the middle represents the median and the upper and lower limits represent the maximum and minimum values. Two-tailed Mann–Whitney tests. (Left) $p = 0.21$, (Middle) **$p:0.0076$, (Right) ****$p < 0.0001$ (**c**) Positions of 3'mRNAseq read clusters along *igdcc3* in 24 hpf sibling null embryo RNA samples, indicating sites of cleavage and polyadenylation. Peak height indicates relative usage of each cleavage site among transcripts of that sample. Dashed lines demarcate intron-2. Black arrows indicate peaks targeted by 3'RACE. **d** 3'RACE

validation of *igdcc3* 3'mRNAseq results in (**c**) in null neurites. Sequencing of products shows terminal intronic sequences containing PAS consensus sequences (red boxes) and the start of a PolyA tail. (Left) exons 1–2 + 6.5 kb intron-2 and polyA. (Right) exons 1–2 + 51 kb intron-2 and polyA. **e, f** Confocal z-projections (10 μm) of 1dpf (1 day post-fertilisation/24hpf) *sfpq* sibling (**e**) & null (**f**) embryo axons, and *igdcc3* IR and PreT-IR transcripts. Probe sets target either the 5'-most (red) or 3'-most (magenta) 10 kb of intron-2. Axons running longitudinally just anterior to the otic vesicle. Asterisks: axonal RNAs labelled by both intron-2 probe sets. Arrows: axonal RNAs labelled by 5' intron-2 probe set only. RNA puncta are also present in neuronal cell bodies surrounding axons. Scale bars 10 μm. **g, h** Quantification of 5' and 3' intron-2 puncta in axons in (**e, f**). Each datapoint represents a different embryo. $N = 10$ sibling and 10 null embryos. Two-tailed unpaired t tests with Welch's correction, ****$p < 0.0001$. Means plotted with SD. Source data are provided as a Source Data file.

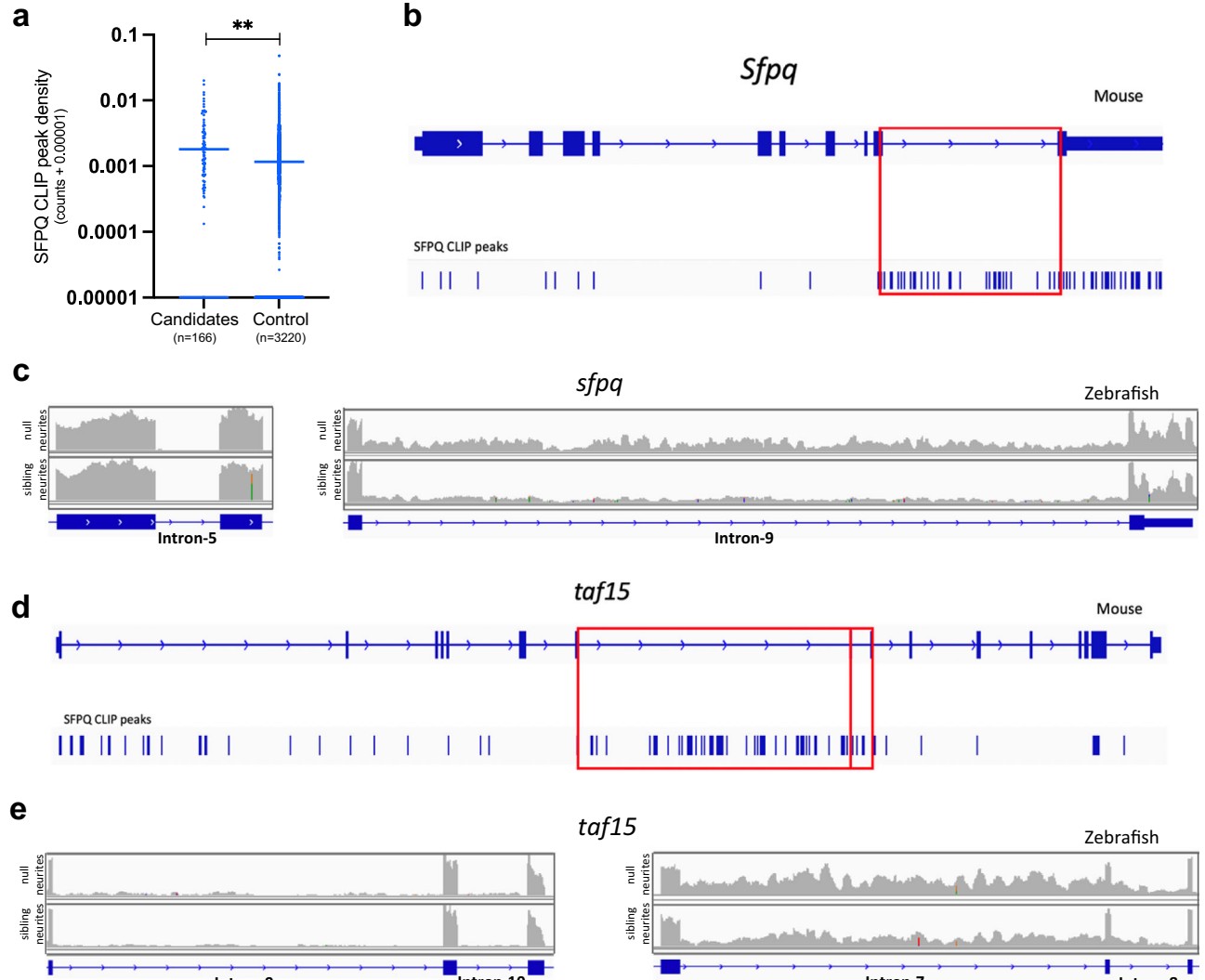

**Fig. 5 | ALS-linked intron retention events are enriched in null neurites. a** CLIP analysis showing the extent of SFPQ binding to mouse homologues of ALS-associated introns. CLIP peak density scores normalise for intron length. Control introns are those not retained in the same genes as ALS-associated introns. Two-tailed Mann–Whitney test, **$p < 0.0057$. **b** Intronic CLIP peaks in *Sfpq*, most

mapping to intron-9 (red box). **c** *sfpq* RNAseq reads of control intron-5 (Left) and retained intron−9 (Right) in sibling and null neurite samples. **d** CLIP peaks mapping to *taf15*. The vast majority of peaks that map to intronic sequences do so in intron-7 and intron-8 (red boxes). **e** *taf15* RNAseq reads of control introns −9 & −10 (Left) and retained introns −7 & −8 (Right) in sibling and null neurite samples.

were validated by 3'RACE for two highly expressed candidates, *ebf1a* (Fig. 6a) and *cux1a* (Fig. 6b). In both cases, these PreT-IR mRNAs have in-frame termination codons positioned early in the intron, strongly suggesting the local production of truncated proteins. We

further validated the presence of *ebf1a* intron-6 PreT-IR transcripts in axons in 24hpf *sfpq* sibling versus null axons by HCR (Fig. 6c–f).

This analysis reveals that IR and PreT-IR transcripts found in neurites of SFPQ-depleted neurons are shared by ALS-patient tissues

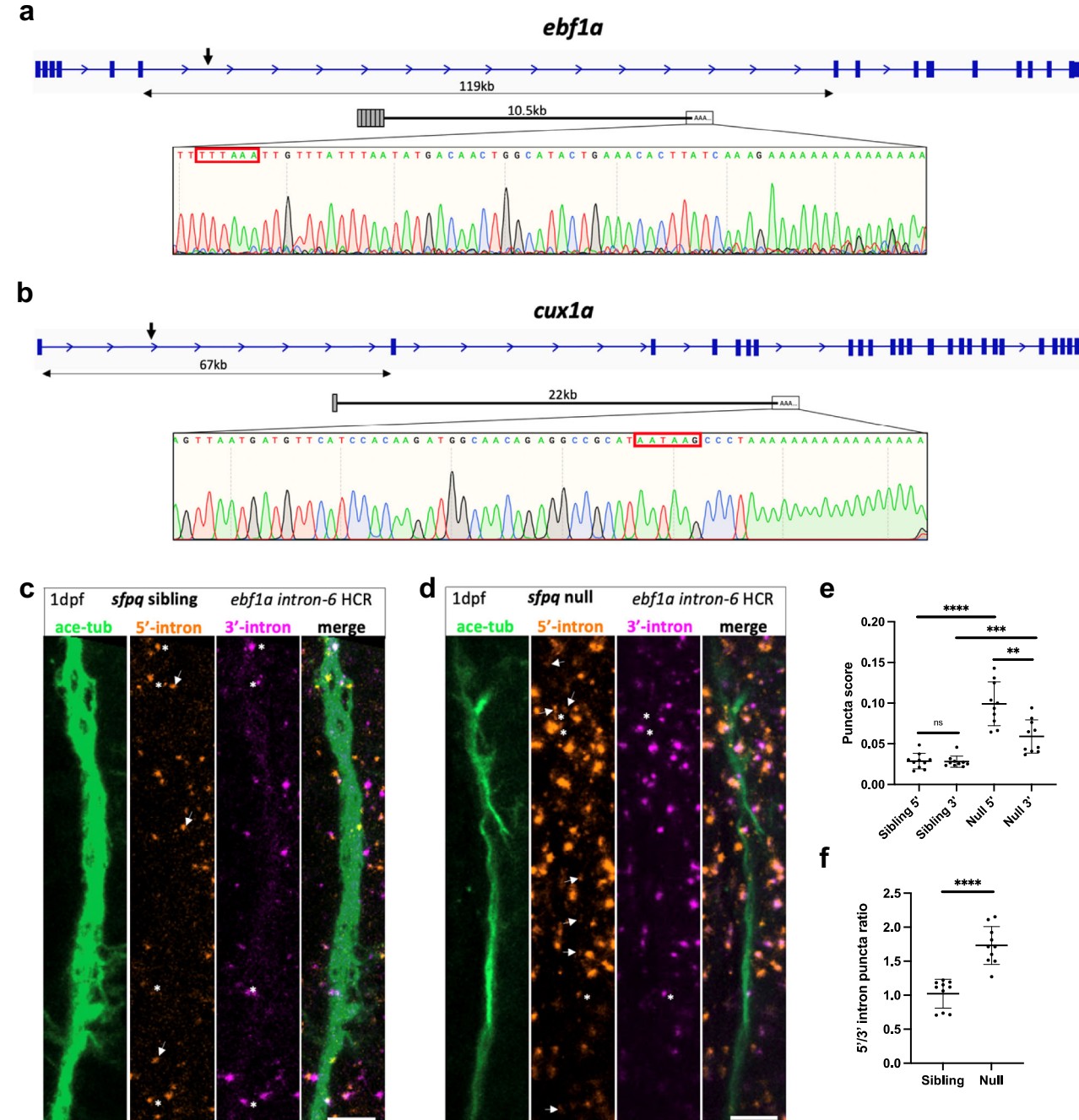

**Fig. 6 | ALS PreT-IR transcripts are enriched in null neurites. a, b** Sequencing traces from 3'RACE products of PreT-IR transcripts for *ebf1a* (**a**) and *cux1a* (**b**), zebrafish homologues of ALS-affected *EBF1* and *CUX1*, show terminal intronic sequences with polyadenylation signals (red boxes) and start of the polyA tail. **a** *ebf1a* PreT-IR transcript comprises exons 1–6 + 10.5 kb intron-6 and polyA tail. **b** *cux1a* PreT-IR transcript comprises exon 1 + 22 kb intron-1 and polyA tail. **c, d** Confocal z-projections (10 µm) of 1dpf/24hpf *sfpq* sibling (**c**) & null (**d**) embryo axons, immunolabelled by targeting acetylated-tubulin, and *ebf1a* intron-6 IR and PreT-IR transcripts labelled by HCR with probe sets targeting the 5'-most 10 kb of intron-6 (red) and the 3'-most 10 kb of intron-6 (magenta). Axons running longitudinally, situated just anterior to the otic vesicle, were imaged in each embryo. Asterisks show axon localised RNAs labelled by both 5' and 3' intron-6 probe sets. Arrows show axon localised RNAs labelled by 5' intron-6 probe sets only. Note that HCR RNA puncta are also present in neuronal cell bodies surrounding shown axons. Scale bars 10 µm. **e, f** Quantification of 5' and 3' intron-6 puncta in sibling (**c**) versus null (**d**) axons. Each datapoint represents a different embryo. $N = 10$ sibling and 10 null embryos. Two-tailed unpaired t tests with Welch's correction, **$p$:0.0016, ***$p$:0.0008 ****$p < 0.0001$. Means plotted with SD. Source data are provided as a Source Data file.

and cellular models, strengthening the functional link between loss of SFPQ function and ALS. Our data indicates that PreT-IR is a novel disease hallmark and likely neurite-centred player in motor pathology[41].

## Discussion

In this work we investigated the function of SFPQ in neurons, focusing specifically upon its local impact in neurites. Previous studies have reported that the cytoplasmic pool of SFPQ is crucial for axonal

development, at least in part through its control of local mRNA expression levels[4,8,9]. Roles in RNA transport, and local translation through its association with late endosomes have been explored thus far[42]. SFPQ loss-of-function is increasingly associated with ALS/FTLD, hypothesised to be a responsible factor that causes neuronal degeneration underpinning pathology.

We found that developing motor neurons cell-autonomously require SFPQ for normal axon extension, branching and synaptogenesis. Motor neurons lacking SFPQ degenerate by exhibiting a "dying back" phenotype reminiscent of ALS. Exploring the impact of SFPQ absence on the subcellular transcriptome, we observed neurite-specific decreases in neuronal mRNAs and neurite-specific increases in IR transcripts. The aberrantly retained introns, rich in SFPQ-binding sites, are abundant in ALS samples, consistent with reports suggesting that SFPQ misregulation is central to the disease. Unexpectedly, many of the abnormal long IR mRNAs are prematurely terminated (PreT-IR), polyadenylated at intronic polyadenylation sites (iPAS), exported from the nucleus, and are subsequently either targeted to neurites or are selectively protected from degradation in neurites. We find these PreT-IR transcripts in ALS patient samples, identifying them as a novel ALS hallmark. Together, these data suggest that loss of SFPQ function leads to 'local' neurite PreT-IR transcript related pathology in ALS/FTLD.

While loss of SFPQ is known to cause aberrant alternative splicing, ours is the first study to show specific impact on local transcriptomes. Axonal and dendritic changes in alternatively spliced isoforms likely have more profound consequences upon the neurite proteome compared to mRNA expression level changes, which may be relatively well tolerated by increasing the local translation of fewer mRNA species. Extensive localisation of classic IR transcripts to neurites observed in absence of SFPQ is surprising, as it is expected to result from changes in nuclear alternative splicing. This predominant increase in neurites suggests that IR transcripts are actively targeted to axons and dendrites or are stabilised there. Given the many possible roles that introns can serve in the cytoplasm, including in RNA transport, degradation, or in diversifying the protein pool, aberrant null neurite-localised IR-transcripts likely perturb neurite homeostasis, contributing significantly to motor axon defects observed in vivo.

Stabilisation of 3'-truncated transcripts by alternative cleavage and polyadenylation, occurring mostly within the retained intron, indicates that SFPQ prevents iPAS usage within long introns. This control is likely direct, owing to the abundance of SFPQ binding in these introns, as well as the lack of expression change in the null of factors that activate or repress iPAS usage (e.g., *CSTF64, FIP1, RBP6/Iso3, PCF11, PABPN1*, *CF* cleavage proteins). Intronic polyadenylation is an emerging form of polyadenylation, frequently associated with immune cells and cancer, where it has been shown to produce truncated pathogenic peptides[31,32,43]. Our findings are the first linking iPAS usage to mislocalisation of PreT-IR transcripts to neurites in neurodegenerative models and support the hypothesis that iPAS misregulation may serve as a universal hallmark of degenerative disorders with motor phenotypes[41]. Stabilised partial mRNA localisation to neurites suggests the formation of local interfering truncated peptides as a mechanism of triggering neurodegeneration.

Misregulation of SFPQ, TDP-43 and FUS, are established hallmarks of ALS, with TDP43 affected in almost all cases[19,44–46]. The regulation of very long introns by such ALS-linked RBPs is also increasingly reported as disrupted[47–50]. While both TDP43 and FUS binding is enriched in very long introns, limited evidence has supported their co-regulation of common introns[51–53]. Interaction of SFPQ with FUS is relatively well established[6,8], while interaction with TDP-43 in neurons is less clear[54,55]. Misregulation of TDP-43, FUS or SFPQ may converge on regulation of common long introns, resulting in distinct splicing defects but causing a common degenerative phenotype.

Numerous reports across genetic (including *FUS* and *SOD1* mutation) and sporadic ALS/FTLD cases indicate that various initial molecular dysfunctions converge on SFPQ function[4,6,7,18–20,56]. In light of our findings, we propose that perturbed control of intron retention in ALS includes the formation of pathogenic PreT-IR transcripts induced by loss of SFPQ function in both the nucleus and neurites, and that this misregulation impacts patients' neurites most severely. Supporting this model, we found that the intron retention signature shared across ALS studies is also present in zebrafish null neurons[18,19,33,34]. Loss of SFPQ function likely perturbs the splicing of SFPQ-binding introns. Indeed, it has been reported that the protein is an essential component, alongside FUS, of an intron-binding complex essential for alternative splicing events which are crucial in preventing tauopathy and an FTLD-like phenotype[6,7]. Most significantly in support of our model, we identify long gene transcriptopathy coupled with misregulated iPAS usage as a new ALS hallmark. The PreT-IR transcripts produced are most abundant in neurites. These mRNAs likely localise to axons and dendrites in patients as they do in null neurons.

The extent of the contribution of perturbed intron retention and long gene transcriptopathy to ALS pathogenesis is yet to be determined. Recently, ALS-associated increases in intron retention were suggested to be a cytoplasmic phenomenon[57]. Our study is the first to indicate that ALS-linked increases in IR and PreT-IR transcripts are neurite-specific, improving our understanding of the molecular pathology. Aberrant pools of such transcripts may affect neurites through generating toxic protein isoforms, promoting aggregation, competing for mRNA regulation, or impacting local miRNA function.

The formation of cryptic last exons (CLEs) is now often found as splicing defect in degenerative disorders[48–50], and we previously reported CLEs in *sfpq* mutant embryos[47]. CLE transcripts are formed by use of a cryptic exon inside annotated introns. In the *sfpq* mutant, the introns affected by CLEs and IR/PreT-IR are largely distinct. There is a very small number of introns forming CLEs in 24 hpf whole *sfpq* embryos that are PreT-IR in our neurite-specific dataset. Overall, neurons are more prone to IR/PreT-IR than CLEs, probably because they express a much higher number of transcripts from genes containing very long introns. The predominant impact of IR/PreT-IR is local (in axons and dendrites). It remains unclear whether this is also the case for CLEs.

Transcripts coding for splicing proteins are disproportionately affected by intron retention in ALS[19,33], further suggesting perturbed local RNA processing, particularly intron-mediated local processing. Overall, the changes that occur to the neurite transcriptomic landscape in absence of SFPQ likely profoundly impact the neurite proteome, and subsequently, axon and dendrite development and synaptic signalling. Therefore, future studies aimed at restoring SFPQ-dependent control of intron retention and iPAS usage may be game-changing in the development of therapeutics for tackling neurodegeneration.

## Methods

### Fish husbandry and transgenic lines

All fish in this study were reared on a 14-hour light/10-hour dark cycle at the Guy's Campus Zebrafish Facility, King's College London. Embryos were maintained at 28.5 °C in either fish system water with 0.01% methylene blue, or E2 solution with 1X Sigma–Aldrich Penicillin-Streptomycin (P4333). Embryos were transferred into clean fish water/E2 medium daily. From 1–5 dpf, embryos were also incubated with 0.003% 1-phenyl-2-thiourea (PTU) to minimise pigmentation.

The *sfpq*, tg(mnx1:GFP) line contains the *sfpq*[kg41] (C-to-T) mutation[4]. Null mutants were identified by phenotype at 24 hpf. Work was approved by the local Animal Care and Use Committee at King's College London and under a license issued by the Home Office. Experiments were performed following the Animals (Scientific Procedures) Act, 1986.

## Cell transplantation and donor embryo identification

Transplant experiments were performed as described previously[4] when embryos were between 50% epiboly and shield stage. A single donor embryo of the *sfpq*, tg(mnx1:GFP) line and three wild-type AB host embryos were lined up in a single vertical strip on the right side of a glass coverslip (donor at the top). Approximately 60 cells were taken from the donor from the region committed to giving rise to ventral spinal cord tissue. An equal number of cells was distributed to each of three wild-type embryos, targeted to the equivalent region from which they were taken to achieve spinal cord labelling. The coverslip with 4 embryos was transferred into a 35 mm petri dish and incubated overnight at 28.5 °C in E3 medium containing 1X Penicillin-Streptomycin (Sigma–Aldrich, P4333).

The identity of donor motor neurons in each host was determined by observing the phenotype of the donor in each transplant dish at 24 hpf. Donor health was sometimes compromised by heavy manipulation during transplantation. Consequently, all were subsequently genotyped by PCR sequencing. Embryos were incubated with 45 µl alkaline lysis buffer (25 mM NaOH, 0.2 mM EDTA) for 1-hour at 95 °C, after which 50 µl neutralisation buffer (40 mM Tris-HCl, pH8.0) was added. 1 µl was used for PCR, amplifying a 645 bp fragment of *sfpq* containing the mutation site (forward primer: GACCGATCTCGTAATCGTGGAGG, reverse primer: CTGGGGCTGA-GAGCTCTGCTCG). Samples were purified using a QIAquick PCR purification kit (QIAGEN, 28104), and sent off for Sequencing (primer: CTCCGCCGAAAATCCAGTCT).

## Zebrafish primary cell culture

From fertilisation, *sfpq*, tg(mnx1:GFP) embryos were incubated in autoclaved fish water containing 0.01% methylene blue (AFW), 1X Penicillin-Streptomycin (Sigma–Aldrich, P4333) until 24 hpf before bleaching twice with 0.0026% sodium hypochlorite (Fisher Scientific, S/5042/15), spending 5-minutes in fresh AFW between each incubation. Following dechorionation with Pronase (Sigma–Aldrich, 10165921001) and sorting into siblings and nulls based on phenotype, embryos were incubated with 1 ml PBS and put on ice before moving to tissue culture for dissociation.

Embryos were transferred into a sterile 40 mm cell strainer (Corning, 431750), which was dipped into 70% ethanol in autoclaved Milli-Q water for 5-seconds. The cell strainer was immediately immersed in Leibovitz L-15 medium (Gibco, 11415049) to remove ethanol. Sterilised embryos were immersed in 1 ml of cell dissociation solution (25% Cell Dissociation Buffer (enzyme free), PBS & $3.25 \times 10^{-3}$M EDTA, pH 8.0) before settling on ice.

Each group of embryos was dissociated by passing them rigorously up and down through autoclaved cotton plugged Pasteur pipets (150 mm and 230 mm; Fisherbrand, 13-678-8 A, 13-678-8B) with successively smaller bore diameters made using a Bunsen burner. PVC teats were sterilised with 70% ethanol in autoclaved Milli-Q water. After approximately 15-minutes, when chunks of embryos were no longer visible, the solution was passed through a Corning 40 mm cell strainer (431750), filtering dissociated cells into a 50 ml STARLAB Falcon centrifuge tube. Cells were centrifuged at 300 g for 7-minutes at 4 °C and resuspended in Zebrafish Neural Medium (ZNM) consisting of Leibovitz L-15 medium (Gibco, 11415049), 1X N-2 Supplement (Gibco, 17502048), 1X MACS NeuroBrew-21 Supplement (MACS, Miltenyi Biotec, 130-093-566), 10 ng/ml BDNF (Sigma–Aldrich, B3795), 2% FBS, 1X Penicillin-Streptomycin (Sigma–Aldrich, P4333) & 50 mg/ml Gentamicin (Gibco, 15750060). The cell solution was subsequently plated onto 12 mm Poly-D-Lysine/Laminin-coated coverslips (Corning, 354087) in a 24-well-plate, or onto cell culture transwell inserts, 1.0 mm PET (Millicell, MCRP06H48). Inserts were pre-coated with 10 mg/ml Poly-D-Lysine (Sigma–Aldrich, A-003-E) on the upper and lower membranes for 2-hours at RT, and 3 mg/ml Laminin Protein (Gibco, 23017015) for 1-hour on the lower membrane. A 1X PBS wash was performed. Plates

were wrapped with parafilm to seal and left overnight at RT. Although cells of every tissue type were plated following whole embryo dissociation, cultures become increasingly neuronal given the nature of the medium that they are in (Supplementary Figs. 2, 3). The upper surface, on which somas and a significant amount of neurite tissue sit, is referred to as the 'cellular' compartment. The lower surface is referred to as the 'neurite' compartment given that an insignificant amount of other cellular extensions passes through 1 µm pores.

## Immunohistochemistry and synapse labelling

24–72 hpf embryos were fixed overnight with 4%-paraformaldehyde (PFA) at 4 °C, then washed three times for 5-minute with PBS, 1%-Triton X-100 (PBST). Incubation with 0.25% Trypsin was carried out for 3-minutes on 24 hpf embryos, and 10-minutes on 72 hpf embryos. The reaction was stopped with heat-inactivated goat serum (HINGS), and 3 × 5-minute PBST washes. Blocking with 10% HINGS in PBST for 1-hour at RT was followed by incubation with primary antibodies in 10% HINGS in PBST overnight at 4 °C. At least 3 X PBST washes over 4–6 h were performed at RT. Secondary antibodies and 1:100 α-Bungarotoxin/α-BTX-555 (Thermo Fisher Scientific, B35451) conjugate to label Acetylcholine receptors in the post-synapse of neuromuscular junctions were incubated in 10% HINGS in PBST overnight at 4 °C, or for 2-hours at RT. Following secondary antibody and α-BTX-555 removal, at least 3 X PBST washes were performed over 4 to 6-hours at RT. Embryos were stored in 70% glycerol.

Immunolabelling on primary cultured cells involved fixing cells at DIV1–3 with 4% PFA was added to each well for 30-minutes at RT. Coverslips underwent 2 × 5-minute PBS-0.25%Triton (PBST0.25) washes before incubation with blocking solution (PBST0.25 and 5% Goat Serum) for 1-hour at RT, and then fresh blocking solution containing primary antibodies overnight at 4 °C. Coverslips were subsequently washed 3 times with PBST0.25 at RT whilst gently shaking. Fresh blocking solution was added to them containing secondary antibodies, and DAPI (1:2000). Coverslips were incubated either at RT for 2-hours or overnight at 4 °C. This solution was then removed, and coverslips were washed three times with PBST0.25 for a few hours. Coverslips were mounted onto SuperFrost+ Adhesive slides (Menzel Thermo Scientific, J1800AMNZ) with FluorSave Reagent (Calbiochem, 345789).

Primary antibodies: 1:500 rabbit anti-GFP (Torrey Pines Biolabs Inc, TP401), 1:100 mouse anti-SV2A (Developmental Studies Hybridoma Bank, AB2315387), 1:1000 mouse IgG2b anti-acetylated tubulin (Sigma–Aldrich, T6793).

Secondary antibodies: All used at 1:1000: Anti-rabbit IgG (H + L) Alexa Fluor 488 (Invitrogen, A11008), Anti-mouse IgG (H + L) Alexa Fluor 488 (Invitrogen, A11001), Anti-mouse IgG (H + L) Alexa Fluor 568 (Invitrogen, A11004), Anti-mouse IgG (H + L) Alexa Fluor 633 (Invitrogen, A21050).

## In situ hybridisation

**RNAscope.** The RNAscope Fluorescent Multiplex Assay v1 was performed on zebrafish primary cells cultured on 12 mm BioCoat Poly-D-Lysine/Laminin-coated coverslips (Corning, 354087), following the RNAscope Fluorescent Assay for Cultured Adherent Cells with some key modifications made. Cells incubating with ZNM had this removed and underwent 1X wash with PBS, before fixation with 1 ml of 4% PFA for 30-minutes at RT and then two further washes with PBS. Coverslips were immersed in 50% ethanol (Sigma–Aldrich, 32221-M) in Milli-Q water, then 75% and finally 100%. The coverslips were incubated in this overnight at −20 °C. Coverslips were rehydrated using the same sequence of immersions in reverse and then washed with PBST for 10-minutes. This was removed and a 15-minute incubation was performed with Protease III (1:15 dilution) at RT before 2X PBS washes. Probes were warmed and mixed as required. Coverslips were placed face down on 25 ml droplets of probe solution and incubated at 40 °C for

2-hours. 2 × 5-minute washes were performed at RT with RNAscope Wash Buffer before overnight incubation at RT in 5X saline-sodium citrate (SSC) buffer. Incubations were then performed, in the same manner, with AMP1 (30-minutes), AMP2 (15-minutes), AMP3 (30-minutes) & AMP4-Alt A, B or C (15-minutes) – all at 40 °C. Between each of these incubations, and after the final incubation with AMP4, coverslips were transferred back to the 24-well-plate where 2 × 5-minute washes were performed using RNAscope Wash Buffer.

Immunohistochemistry was performed following RNAscope as described above, incubating coverslips first with PBST (0.25%) for 20-minutes, before 90-minutes with PBST & 5% goat serum. Following immunostaining, coverslips were incubated with DAPI (Sigma-D8417) at 1:2500. Following a final PBS wash, coverslips were mounted for imaging as described above. RNAscope punctae observed in neurites were quantified manually, scoring puncta based on fluorescent intensity to infer the number of transcripts per granule.

Probes were designed by, and purchased from, Advanced Cell Diagnostics (Dr-*gria4b*-intron-2 5', catalogue number: 585881; Dr-*nova2*, catalogue number: 585921; RNAscope Negative Control Probe – DapB, catalogue number: 310043).

**Hybridization chain reaction.** Probe sets consisted of the following: *gfp* (11 probe pairs)*, igdcc3*-intron-2 5' (18 probe pairs), *igdcc3*-intron-2 3' (18 probe pairs), *ebf1a*-intron-6 5' (21 probe pairs), *ebf1a*-intron-6 3' (21 probe pairs). The whole-mount embryo in situ hybridization protocol was followed as described[58]. Following completion of HCR, immunohistochemistry was performed targeting axons using acetylated-tubulin antibody according to the protocol[58]. Embryos were mounted anterior to the left, and longitudinal axon tracts just anterior to the otic vesicle were imaged. 10 μm z-projections were prepared and axon-localised intron puncta were quantified manually. Puncta score was calculated by normalising the number of puncta counted in an axon to the total axon length.

**Imaging.** A Nikon D-Eclipse C1 microscope with 488 nm and 543 nm lasers using a Nikon Fluor 40x (NA: 0.80, Water) objective was used for confocal live-imaging of transplanted GFP-positive motor neuron development between 1–5 dpf (z-projection step size of 1.5 μm). For live imaging, each day, embryos were anaesthetized with 0.02% MS222 and mounted in 1% low melting point agarose. Imaging was performed every 24-hours. Any fixed & immunolabelled embryos were manually deyolked and mounted for imaging in 70% glycerol in PBS.

A Zeiss Axio Imager.Z2 LSM 800 confocal microscope with Airyscan equipped with 2x GaAsP spectral detectors was used for imaging of fixed whole embryo SV2 antibody, α-BTX-555 & HCR probe labelling, and fixed primary culture. Excitation was provided by 405 nm, 488 nm, 561 nm and 633 nm solid-state lasers. Images were captured using 40x (NA: 1.3, Oil) or 63x (NA: 1.4 Oil) Zeiss objectives at 1024 × 1024-pixel resolution.

**Motor axon branching and synapse analysis.** Microscope images were processed and analyzed using Fiji. Transplanted GFP-positive motor neuron branches and clone numbers were quantified manually. The dorsoventral position of cell somas within the spinal cord was calculated by measuring the distance from the border of the notochord and spinal cord to the center of the cell soma. Cell soma diameters were measured at their widest point. Distal portions of ventrally projecting motor axons were selected for quantification in our synaptogenesis analysis. The amount of axon colocalisation with SV2 and α-BTX was calculated by dividing the length of cumulative colocalizing axon over the total axon length in the image. RNAscope puncta in neurites were quantified by manual scoring.

**Tissue and total RNA extraction from transwell inserts and RNAseq.** All surfaces and instruments were decontaminated with RNase Zap

(Invitrogen, AM9780). At DIV2, ZNM was pipetted from above and below the membrane of transwell inserts with cells incubating in them. 1X PBS wash was performed and then pipetted away before the insert was removed from the well. A microdenier sterile swab (Texwipe, TX761MD) was used to wipe as much material as possible, quickly but gently, from the cellular/upper membrane surface. The swab head was immersed in an Eppendorf tube containing 350 ml Buffer RLT buffer from the Qiagen RNeasy Micro Kit (74005). A cell scraper (Nunc, 179693) was used to remove material from the lower surface of the transwell insert. The scraper head was then dipped into 350 ml Buffer RLT buffer from the Qiagen RNeasy Micro Kit (74005). Swab heads were removed after a few minutes of soaking. Extra effort was taken to squeeze Buffer RLT and cellular tissue from the swab before it was discarded.

The RNeasy Micro Kit (Qiagen, 74005) protocol was carried out, following minor modifications, on all samples. To ensure the removal of all traces of Buffer RW1, tubes were slowly rotated following the addition of 500 ml Buffer RPE to each column. After centrifugation, this step was repeated, with 500 ml Buffer RPE added to the tube, rotation, and centrifugation. RNA samples were stored at −70 °C. Concentrations were measured by Qubit RNA HS Kit (Invitrogen, Q32852) using 2 ml of each sample. Typical concentrations for extracted RNA: sibling cellular (90 ng/μl), sibling neurite (20 ng/μl), null cellular (50 ng/μl) and null neurite (10 ng/μl). RNA quality of all samples was assessed at the King's Genomic Centre, by Agilent 2100 Bioanalyzer. 11 of the 12 samples showed RIN > 8.5.

RNA samples were sent on dry ice to the Huntsman Cancer Institute High-Throughput Genomics Facility, University of Utah, USA, for all library preparation and sequencing steps. The TruSeq Stranded Total RNA Library Prep Gold kit was used for library preparation, removing ribosomal RNA by Ribo-Zero Gold depletion (Illumina, 20020598). Samples were applied to an Illumina NovaSeq flow cell and sequenced using the Illumina NovaSeq 6000 instrument. 100 million 50-base pair read-pairs were purchased for each of the 4 sample types.

**Reverse transcription-quantitative PCR (RT-qPCR).** Following Total RNA extraction from cellular and neurite tissue isolated from transwell insert cultures, cDNA synthesis was performed using the First Strand cDNA synthesis kit (Thermo Scientific, K1612) using random hexamer primer. Control cDNA synthesis reactions, in which no reverse transcriptase was added were run alongside samples to rule out gDNA contamination for detection of intronic sequence in sample cDNA. RT-qPCR reactions were performed using LightCycler 480 SYBR Green I Master kit (Roche, 04707516001), run on a LightCycler 96 (Roche, 05815916001) system. Analysis was performed using LightCycler96 software. All RT-qPCR reactions were normalised to the expression of, *actl6a*, calculating ΔCt. For the analysis of changes in gene expression, null sample values were made relative to sibling samples of the equivalent compartment−cellular or neurite−giving ΔΔCt, allowing calculation of relative expression ($2^{-\Delta\Delta Ct}$).

For intron retention validation, intron-exon amplicons were made relative (ΔΔCt) to either their respective compartment exon fragment expression for the gene or the expression of the intron-retaining fragment of the other compartment of the same genotype (i.e., sibling neurite relative to sibling cellular). Control fragments, of introns not retained, were also made relative to the expression of the retained intron-exon fragment.

In our validation, we observed retained intron expression to be consistently more abundant in neurites than in the cellular compartment of null samples, eliminating the possibility that detection of introns in neurites may reflect pre-mRNA bleedthrough from the cellular compartment.

**3'-RACE.** Tissue was isolated and total RNA extracted from null cells incubating on transwell inserts at DIV2 as described above. First-strand

cDNA synthesis was performed on null cellular, null neurite, and control RNA samples using the 3′ RACE System for Rapid Amplification of cDNA Ends with the adapter primer (Invitrogen, 18373019). Both the first and second rounds of cDNA amplification used the AUAP primer as reverse primer. GSP and nested forward primers used for the various reactions are listed in Supplementary data 10. Amplification-1 included 35 PCR cycles. Amplification-2 included 40 PCR cycles. Amplicons from the second reaction were purified using the MinElute PCR Purification Kit (QIAGEN, 28004) and sent off for DNA sequencing.

**Compiling ALS-associated introns.** 201 human and mouse introns were identified from various studies (Supplementary data 8). 80 human introns were sourced from a single study; their retention increased across multiple ALS datasets[19]. A study comparing RNAseq datasets from *Fus* knockout and *Fus* ALS-model mice identified 5 introns consistently perturbed in ALS models only, and a further 96 introns consistently perturbed in the ALS model as well as knockout datasets[33]. We also performed our analysis of four sporadic ALS patient tissue RNAseq datasets, identifying an additional 20 human introns (from 11 genes) affected in two or more of the four studies[35–38]. The candidate intron affected in each gene sometimes differed between studies. All 201 of these introns were inputted into LiftOver, identifying 145 Zebrafish homologous introns (GRCz10 assembly), which we then visualised in our dataset using Integrated Genome Viewer (IGV)[59].

**Bioinformatics.** The quality of sequencing reads (FASTQ) was checked using FastQC[60], and high-quality sequences were selected using Trimmomatic's[61] sliding window approach. Read alignment to GRCz10 transcriptome and gene expression quantification was performed using VAST-TOOLS'[62] 'align' function as follows:

vast-tools align read1.fastq.gz read2.fastq.gz --sp Dre --cores <num_threads >.

The above function outputs gene-level count matrices. Differential gene expression analysis was performed in R using the edgeR package[63] with the estimateGLMRobustDisp model. Genes with low expression (<20 transcripts per million (TpM)) in sibling neurite samples were filtered from analyses.

Differential splicing analysis was performed using VAST-TOOLS' 'diff' function as follows:

vast-tools diff -i vasttools.tab -a sampleA_replicates -b sampleB_replicates -m 0.01 -c 40 -e 5.

Splicing events were deemed significantly changing if the variable "MV d[PSI]_at_0.95" is above 0. Details on thresholds for the analyses described in the results are outlined in the relevant Supplementary data legends.

In addition to VAST-TOOLS, intron retention analysis was performed using IRFinder[28]. Sequencing reads from the three biological replicates for each sample were merged to provide sufficient depth for comparative analysis. IRFinder was performed according to the developer's recommendation and significantly changing events were selected using a *p*-value cut-off of <0.05.

To quantify the distribution of reads along genes, we first mapped high-quality sequences to GRCz10 genome using HISAT2[64] as follows:

hisat2 -p <num_threads > -x < GRCz10 index > −1 read1.fastq −2 read2.fastq -S output.sam.

HISAT2-mapped reads were converted to BAM format using SAMtools[65]. A BED file containing coordinates of binned spliced genes was generated using the R package GenomicRanges. Essentially, GRCz10 reference transcriptome (GTF) was imported into R and genes shorter than 50 bp were filtered out. Genes with redundant exons, due to the presence of alternative isoforms, were simplified using GenomicRanges' 'reduce' function and the entire set of genes were binned into 50 equal parts (features) using custom R scripts. The number of reads mapping to each feature was quantified using Bedtools'

coverageBed function. Genes with total coverage of fewer than 50 reads were filtered from the analysis. Counts for each feature were normalised to the total coverage of its gene and a linear regression model for each gene was constructed using bin number as the dependent variable and normalised read counts as the independent variable. The slope coefficient ($m$) for each gene was extracted from the models and delta.slope was calculated by normalizing the slope($m$) in null neurite samples to the slope ($m$) in sibling neurite samples.

3′mRNA-seq analysis was performed using our previously generated data from RNA extracted from 24 hpf sibling and null embryos[47]. Major clusters of 3′-mRNAseq reads were compared between 'classic' full-length versus 'sloping' genes. Analyses were conducted comparing differences in cluster number along the length of whole genes, as well as within the retained introns. The extent to which premature cleavage sites were used was compared by quantifying the intensity of clusters within the gene and retained introns.

Identification of IR genes whose transcripts undergo translation preferentially in neurites was performed using the interactive web interface (https://public.brain.mpg.de/dashapps/localseq/info)[1] after identifying homologous Rattus norvegicus gene names.

**Reporting summary**
Further information on research design is available in the Nature Research Reporting Summary linked to this article.

## Data availability
RNA sequencing data generated in this study have been deposited in the ArrayExpress database under accession code E-MTAB-11431. 3′mRNA-seq data from *sfpq* sibling and null zebrafish lines[47] is available from ArrayExpress at accession code E-MTAB-9899. Mouse Sfpq CLIP-seq data[22] is available at accession number GSE60246. GRCz10 assembly is available for download at link (https://www.ncbi.nlm.nih.gov/assembly/GCF_000002035.5/). All data generated in this study are provided in the Supplementary Information/Source Data file. Supplementary Tables 1 and 2 are in the Supplementary Information. Source data are provided with this paper.

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

## Acknowledgements

We thank the King's College London Guy's campus fish facility staff for their fish husbandry and care. We thank Oscar Marin for critical reading of the manuscript. We thank Brian Dalley at the Huntsman Cancer Institute, University of Utah for conducting RNA sequencing. We thank Simon Bullock, Christine Holt, and members of the Houart lab for stimulating discussions. Biotechnology and Biological Sciences Research Council grant BB/P001599/1 to C.H. Biotechnology and Biological Sciences Research Council grant BB/M007103/1 to E.V.M. Medical Research Council grant MR/T033126/1 to C.H. Medical Research Council studentship MR/K50130X/1/ to R.T. King's College London studentship to R.T. Wellcome Trust Investigator Award WT 220861/Z/20/Z to C.H.

## Author contributions

C.H. conceived the research project and secured funding. R.T. and C.H. designed the experiments. R.T. conducted the experiments. T.F. and P.M.G. provided support performing some experiments. R.T., F.H., M.M. and E.V.M. analysed the data. F.H. performed the bioinformatic analyses. R.T. and C.H. wrote the manuscript with input from the other authors.

## Competing interests

The authors declare no competing interests.
