## [Peer Review File · Nature Communications]

Prematurely terminated intron-retaining mRNAs invade axons in SFPQ null-driven neurodegeneration and are a hallmark of ALSREVIEWER COMMENTS

Reviewer #1 (Remarks to the Author):

Mutation of SFPQ is associated with motor degeneration and maturation deficits that occur in ALS. As a splice regulator, it has been thought that SFPQ mutation leads to aberrant splicing and accumulation of toxic aggregating products that lead to neurodegeneration. In this work, the authors explore the mechanism by which SFPQ loss of function leads to neurodegeneration in vivo and in cell culture models. They provide evidence that loss of SFPQ leads to accumulation of intron-retaining (IR) RNAs, many of which are prematurely terminated and surprisingly are both stable and transported to/enriched in axons and dendrites. This is surprising because the expectation would be that such RNAs would be recognized, retained in the nucleus, and degraded. Further the authors provide evidence that the prematurely terminated IR RNAs are polyadenylated. Thus, the authors conclude that subcellular enrichment of these polyadenylated intron retaining RNAs, could lead to local differences in the proteome, in particular in the axons and dendrites impacted by SFPQ mutations. Evidence to support their model includes functional analyses in zebrafish SFPQ mutant models, enrichment of IR and preT-IR RNAs in brain samples of patients with familial and sporadic ALS, and iPSC derived neurons. Consistent with recent studies of SFPQ function, their RNAseq data indicates that down-regulated genes in the mutant contexts correlates with large intron size. The authors provide evidence that these preT-IRs are enriched in neurites and undergo alternative polyadenylation in intronic polyadenylation sequences, which likely accounts for their stability. Intron polyadenylation has been previously observed in other contexts, including cancer and immune cells but not in neurons or ALS models. Further, the authors compared select ALS genes with retained introns and found that the genes with retained introns were shared between zebrafish neurites, iPSC derived models, and ALS patients with SFPQ mutations and that the retained introns tend to contain SFPQ binding sites.

In summary, the authors link iPAS usage to neurodegeneration and provide a potential mechanism for neurite susceptibility namely, production of truncated mRNAs that are transported/accumulate in neurites where they produce local interfering truncated peptides that cause degeneration. Overall, the manuscript is well written, the data are clearly presented, with appropriate numbers and statistical analyses throughout, and support the main conclusions. In the current work it remains unclear if the truncated and polyadenylated RNAs accumulate or are actively transported to neurites; however, this is for the most part qualified and investigating the mechanism of IR RNA accumulation is beyond the scope of this study which provides new insights into the mechanism and selectivity of SFPQ pathogenesis in ALS and neurodegeneration.

Comments:

- 1) The model is that the IR transcripts lead to truncated toxic/pathological proteins that cause degeneration, but some IR transcripts are also observed in normal conditions. Do these transcripts overlap and/or are the truncated proteins produced from IR transcripts found in WT regulatory or otherwise provide clues about the pathology?
- 2) Related to the model and distinguishing between toxic effects due to the RNAs versus truncation, does blocking translation prevent or ameliorate the degeneration phenotypes?
- 3) Larger font in panels F and G of Figure 1 would be helpful.

Reviewer #2 (Remarks to the Author):

“ALS-associated loss of SFPQ triggers neurodegeneration through neurite localisation of prematurely terminated intron-retaining mRNAs” by Taylor et al reports interesting new phenomena related to mRNA processing in neurons as it might relate to ALS/ Motor Neuron disease. The authors find that disruption to the splicing factor SFPQ leads to a range of aberrant mRNA processing events including the appearance of intron-retaining mRNAs (IR) and prematurely terminated intron retaining mRNAs (preT-IR) localised to neurites (not restricted to cell bodies). The manuscript primarily takes advantage of a cell culturing method using zebrafish neurons (control and *sfpq* mutant) that allows the separate sequencing of mRNAs from cell body and neurite compartments. In addition, RNA sequence databases of ALS models are mined in which these aberrant mRNAs appear common, implying that they may represent a unifying feature of ALS. This premise is supported by sequence analysis of ALS tissue itself. Although much of the data in the manuscript are intriguing and the ideas very novel, many of the core observations need validation (major point 1) and the manuscript contains numerous strong conclusions that are currently overstated (major point 2). However, with some key validation experiments and more circumspect conclusions based on the data to hand, this manuscript should be of great interest and suitable for publication in a general interest journal.

Major point 1 (validation).

There are numerous sequence-based observations that need to be validated in tissue.

1a. The first conclusion “SFPQ-depletion results in neurite-specific localisation of transcripts retaining long SFPQ binding introns” needs to have exemplar mRNA species localisation validated in zebrafish tissue, i.e. in motor axons, by high resolution mRNA localisation procedures. At the very minimum the mRNA examined in cell culture should be assessed in zebrafish tissue.

1b. Whereas there is at least some validation of mRNA localisation/ enrichment to neurites in the first dataset, there is no cell or tissue-level mRNA localisation analysed for any of the following studies, and the primary evidence of mislocalised mRNAs is sequence expression-based. Therefore, the conclusion that “Neurite localised IR mRNAs are frequently prematurely terminated and polyadenylated within the intron” can’t be considered fully supported simply by sequence based analysis of an in vitro based model. There simply has to be validation. This would, again, be most convincing in tissue, i.e. in the available *sfpq* mutant zebrafish axons. Validation in cultured neurites vs cell body would likely be simpler, but less convincing, given the novelty of conclusions made.

1c. Again, the same point stands for the final major section “Neurite IR and PreT-IR transcripts are enriched in ALS samples.” The fundamental sequence analyses here have the genuine potential to be paradigm shifting. However, and again, this does in turn put a major onus on validation. Clearly it would be asking too much to insist on extensive validation of IR and preT-IR transcripts across ALS tissue, but if validation could be carried out on tissue from ALS models, this would be hugely encouraging. If at least some analyses of ALS tissue could be made, this would, of course, be very powerful

Major point 2 (overstatement)

There is a significant amount of unnecessary overstatement in the manuscript. The identification of the mRNA species and validation of their mislocalisation in *sfpq* mutant axons (once major point 1 is addressed) alongside the data that these may represent a previously unknown feature of ALS is perfectly sufficient. There is no need to make further unsupported claims.

2a. The title “ALS-associated loss of SFPQ triggers neurodegeneration through neurite localisation of prematurely terminated intron-retaining mRNAs” is, for example, somewhat misleading in terms of the inference that the preT mRNAs play a causal role in neurodegeneration. They might, but that has not been specifically tested here at all, so this conclusion should not be made. The title, abstract and intro contain a huge amount of overstatement, compared, for example, with the excellent discussion section. It would be much more circumspect to stick to the findings that are, or will be, supported here, which, again, are very interesting, even if assessing causality in contributing to neurodegeneration or

disease remains to be determined.

2b. Statements like “Such findings have established SFPQ as a hallmark of ALS/FTLD” are also overstated and unnecessary in the setup. The data in this manuscript may indeed support the idea that disruption to *sfpq* may lead to a signature molecular dysregulation seen across ALS, but that again, remains to all be joined together. The genetic linkage of *sfpq* mutations to ALS per se is not nearly as widespread as the manuscript implies. Again, amending this can be a very simple editorial change, but essential, in my opinion, to provide a more balanced view of the field.

2c. “This finding debunks a major dogma in RNA biology: prematurely terminated mRNAs are not, as expected, systematically retained in the nucleus and degraded.

Again, statements like this cannot yet be made, given the lack of validation. If major point 1 is addressed fully, then some robust conclusions can be made, but in general the language could be tempered to align better with the data presented.

Minor points

Many of the niche terms could do with being explained in the main text, at least in brief, and some details need clarification.

What are “vast tools”?

What is a clip data set?

What is a transcriptopathy?

“Over this extended time course, we find that in over 30% of cases, null motor neurons do extend axons, with a substantial delay compared to sibling (control; *sfpq*+/+ and +/-) counterparts.” This is confusing. Does this mean that in nearly 70% of cases that motor neurons do not extend axons at all, or that there is no delay in the extension of their axons.

The “slope” analysis could be briefly explained in the narrative

“We found that *sfpq*-intron-9 retention (the most reported in ALS) was significantly increased in null samples, and more so in neurites” Does this imply that *sfpq* is required for the processing of *sfpq* mRNA? Is the *sfpq* mRNA still present in the null mutant, and thus not subject to nonsense mediated decay.

What is iPAS?

Reviewer #3 (Remarks to the Author):

In this manuscript, Taylor et al. reported the observation that the loss of SFPQ in neurons led to an accumulation of intron-containing transcripts in neurite. If true, this result may help understand the role of SFPQ in ALS. However, more analyses are needed to strengthen this conclusion. In addition, while such an association is interesting, in the absence of more mechanistic data, the authors need to tune down many of the interpretation/claims, including the claim in the abstract that “neurite-centred perturbation of alternatively spliced isoforms initiates the neurodegenerative process.”

One of the key new results is that SFPQ affects local transcriptome in the neurite. However, the analyses of the neurite RNA-seq data can be strengthened, including 1) quality control showing that neurite enrichment actually worked, e.g., known neurite-specific transcripts are highly enriched in the neurite sample relative to the cellular samples. 2) scatter plots/volcano plots to show the global trend

that there are more genes downregulated than upregulated. While some analyses have been presented, the text in Fig 1F-G are too small to read. Legends seem to be incorrect. Without more thorough validation and more rigorous analyses, I'm not convinced that SFPQ depletion affects the neurite transcriptome.

The calling of retained introns were not done in a rigorously way. While two tools were used (vast and IRFinder) and score cut-off values were cited, no p-value cut-offs were reported. Also what is the overlap of retained introns found by the two tools? In addition to comparisons of two genotype in the same compartment, neurite-to-cellular comparisons of the same genotype should also be included.

Line 192: "Intronic polyadenylation (IpA) is an uncommon subclass of ApA reported to occur in immune cells and otherwise has only been associated with cancer" – this is not true. IpA is common outside of immune and cancer cells. Similarly line 296-298. Also intron retention/polyadenylation has been linked to neurodegeneration.

Line 246: tune down: "...uncovering neurite PreT-IR as a player in pathology". The results showed only indicate an association.

Line 278-286: I do not see why it is necessary to invoke the controversial theory of local splicing in the neurite. Unlike XBP1 whose splicing is independent of the spliceosome, these retained introns contain the canonical splice sites thus are canonical substrates of the nuclear spliceosome.

The same group reported last year that SFPQ represses cryptic last exons, many of which are found in introns. Please comment on the overlap and discuss possible connections.

Xuebing Wu (expertise: RNA and genomics)

We thank the reviewers for their interest, support, and helpful comments in relation to our manuscript and its publication in Nature Communications. Below we detail the revisions we have made in response to all points raised. We hope that the reviewers find the revised manuscript significantly improved and ready for publication.

Reviewer #1

“Overall, the manuscript is well written, the data are clearly presented, with appropriate numbers and statistical analyses throughout, and support the main conclusions. In the current work it remains unclear if the truncated and polyadenylated RNAs accumulate or are actively transported to neurites; however, this is for the most part qualified and investigating the mechanism of IR RNA accumulation is beyond the scope of this study which provides new insights into the mechanism and selectivity of SFPQ pathogenesis in ALS and neurodegeneration.”

1) The model is that the IR transcripts lead to truncated toxic/pathological proteins that cause degeneration, but some IR transcripts are also observed in normal conditions. Do these transcripts overlap and/or are the truncated proteins produced from IR transcripts found in WT regulatory or otherwise provide clues about the pathology?

Only a third of introns showing retention in null neurites also show some lower level of retention in control neurites (cut off: >5% of total transcripts). The pathologies are potentially both due to excess of untranslated IR mRNA carrying regulatory functions and/or to increase in translated truncated proteins.

2) Related to the model and distinguishing between toxic effects due to the RNAs versus truncation, does blocking translation prevent or ameliorate the degeneration phenotypes?

The degeneration phenotypes observed likely result from the combinatorial toxic effects due to IR RNAs or truncated proteins from many genes. We therefore consider it unlikely that blocking any specific IR RNA from being translated would significantly ameliorate the phenotype.

Global blocking of translation would not be specific to IR isoforms. Impacting all mRNA isoforms would profoundly affect neuronal projection development and function, and therefore would also be very unlikely to ameliorate the degenerative phenotype. We have therefore not found a satisfactory avenue to address this point.

3) Larger font in panels F and G of Figure 1 would be helpful.

This point was also raised by Reviewer #3. Our apologies that the font size in these figure axes was too small. It has been increased.

Reviewer #2:

“Taylor et al reports interesting new phenomena related to mRNA processing in neurons as it might relate to ALS/ Motor Neuron disease.”

“Although much of the data in the manuscript are intriguing and the ideas very novel, many of the core observations need validation (major point 1) and the manuscript contains numerous strong conclusions that are currently overstated (major point 2). However, with

some key validation experiments and more circumspect conclusions based on the data to hand, this manuscript should be of great interest and suitable for publication in a general interest journal.”

Major point 1 (validation):

There are numerous sequence-based observations that need to be validated in tissue.

1a. The first conclusion “SFPQ-depletion results in neurite-specific localisation of transcripts retaining long SFPQ binding introns” needs to have exemplar mRNA species localisation validated in zebrafish tissue, i.e. in motor axons, by high resolution mRNA localisation procedures. At the very minimum the mRNA examined in cell culture should be assessed in zebrafish tissue.

*1b. Whereas there is validation of mRNA localisation/ enrichment to neurites in the first dataset, there is no cell or tissue-level mRNA localisation analysed for any of the following studies, and the primary evidence of mislocalised mRNAs is sequence expression-based. Therefore, the conclusion that “Neurite localised IR mRNAs are frequently prematurely terminated and polyadenylated within the intron” can’t be considered fully supported simply by sequence based analysis of an in vitro based model. [...] validation[...] most convincing in tissue, i.e. in the available *sfpq* mutant zebrafish axons. Validation in cultured neurites vs cell body would likely be simpler, but less convincing, given the novelty of conclusions made.*

We now provide zebrafish tissue level HCR in situ hybridisation validation for two IR & PreT-IR candidates: *igdcc3* (Fig.4E-H; fig. S9G-H’’) and ALS-linked *ebf1a* (Fig.6C-F; fig. S12). Increases in IR and PreT-IR transcripts are quantified in sibling and null axons in whole embryos.

1c. Again, the same point stands for the final major section “Neurite IR and PreT-IR transcripts are enriched in ALS samples.” The fundamental sequence analyses here have the genuine potential to be paradigm shifting. However, and again, this does in turn put a major onus on validation. Clearly it would be asking too much to insist on extensive validation of IR and preT-IR transcripts across ALS tissue, but if validation could be carried out on tissue from ALS models, this would be hugely encouraging. If at least some analyses of ALS tissue could be made, this would, of course, be very powerful.

We agree with the reviewer that our ALS data has the potential to be paradigm shifting. It is not specified which ALS model would be suggested, and none are very strong in recapitulating the physiopathology of human patients, therefore, positive results in those would still not provide a strong support for human relevance.

Validation in human tissue as suggested does not enable timely manuscript turnaround given limited access to ALS samples and therefore is not in the scope of this study. The experiment is high on our priority and will form part of a follow-up study.

We would like to draw the reviewer’s attention to the newly provided zebrafish tissue mRNA detection with one ALS PreT-IR transcript unambiguously accumulating in *sfpq* null neurites (Fig.6C-F; fig. S11; fig. S12). Given the fact that the zebrafish RNAseq results we obtained are now validated *in vivo*, the finding we made from RNAseq datasets from human tissues are very likely to be solid.

Major point 2 (overstatement)

*There is a significant amount of unnecessary overstatement in the manuscript. The identification of the mRNA species and validation of their mislocalisation in *sfpq* mutant axons (once major point 1 is addressed) alongside the data that these may represent a*

previously unknown feature of ALS is perfectly sufficient. There is no need to make further unsupported claims.

2a. The title “ALS-associated loss of SFPQ triggers neurodegeneration through neurite localisation of prematurely terminated intron-retaining mRNAs” is, for example, somewhat misleading in terms of the inference that the preT mRNAs play a causal role in neurodegeneration. They might, but that has not been specifically tested here at all, so this conclusion should not be made. The title, abstract and intro contain a huge amount of overstatement, compared, for example, with the excellent discussion section. It would be much more circumspect to stick to the findings that are, or will be, supported here, which, again, are very interesting, even if assessing causality in contributing to neurodegeneration or disease remains to be determined.

We acknowledge these comments and have modified the manuscript title, abstract and discussion sections accordingly.

2b. Statements like “Such findings have established SFPQ as a hallmark of ALS/FTLD” are also overstated and unnecessary in the setup. The data in this manuscript may indeed support the idea that disruption to sfpq may lead to a signature molecular dysregulation seen across ALS, but that again, remains to all be joined together. The genetic linkage of sfpq mutations to ALS per se is not nearly as widespread as the manuscript implies. Again, amending this can be a very simple editorial change, but essential, in my opinion, to provide a more balanced view of the field.

We respectfully disagree that our statement is overstated. Our manuscript has limited emphasis on the genetic linkage of sfpq mutations to ALS as, indeed, only a couple of cases are linked to sfpq mutations. We refer to the published findings showing that protein is misregulated. SFPQ was initially identified several years ago as an ALS hallmark, independent of genetic cause of disease, with nuclear loss and cytoplasmic accumulation observed in both ALS models and sALS patient tissue (Luisier et al., 2018; *Nature Communications*). Since this initial study, others have described misregulation in a variety of patient tissues and disease models (e.g., Hogan et al., 2021, *Neuropathology and Applied Neurobiology*; Ishigaki et al., 2020, *Brain*; Riku et al., 2022, *Brain*).

2c. “This finding debunks a major dogma in RNA biology: prematurely terminated mRNAs are not, as expected, systematically retained in the nucleus and degraded. Again, statements like this cannot yet be made, given the lack of validation. If major point 1 is addressed fully, then some robust conclusions can be made, but in general the language could be tempered to align better with the data presented.

We have now provided the requested validation in relation to this point (Fig.4; fig. S9G-H”; Fig.6C-F; fig. S12) and tempered the language used in the manuscript.

Minor points

Many of the niche terms could do with being explained in the main text, at least in brief, and some details need clarification. What are “vast tools”? What is a clip data set? What is a transcriptopathy?

VAST-TOOLS is a pipeline that detects and quantifies various types of alternative splicing and analyses their differences between samples. This has now been described in the results section (Line 133).

CLIP (cross-linking and immunoprecipitation) is a widely utilised technique by the field that allows one to determine RNA sequences physically linked to a specific RNA-binding protein. As it is routinely used, this term is not explained in main texts of articles anymore.

Transcriptopathy is defined in the relevant results section of the manuscript (Line 176), and a reference is provided to the study that explores this in detail.

*“Over this extended time course, we find that in over 30% of cases, null motor neurons do not extend axons, with a substantial delay compared to sibling (control; *sfpq*^{+/+} and +/-) counterparts.” This is confusing. Does this mean that in nearly 70% of cases that motor neurons do not extend axons at all, or that there is no delay in the extension of their axons.*

Yes, the majority of motor neurons did not extend axons at any of the time points we looked at. This has now been made clearer in the text.

The “slope” analysis could be briefly explained in the narrative

*“We found that *sfpq*-intron-9 retention (the most reported in ALS) was significantly increased in null samples, and more so in neurites” Does this imply that *sfpq* is required for the processing of *sfpq* mRNA? Is the *sfpq* mRNA still present in the null mutant, and thus not subject to nonsense mediated decay.*

The slope analysis has been briefly explained as requested (Line 184-188).

Although dramatically downregulated in our *sfpq* mutant line, likely owing to NMD upon the pioneer round of translation (Thomas-Jinu et al., 2017, *Neuron*), a significant pool of *sfpq* intron-9-retaining mRNA remains detectable. Increased intron-9-retention and strong SFPQ binding to intron-9 (Fig. 5B) is suggestive of self-autoregulation. This has now been specified in the main text (Lines 239-241).

What is iPAS?

Intronic polyadenylation site. This is stated in the text (Lines 200 & 286) and has also been added to the keywords section. It is a site within an intron where a polyadenylation consensus sequence (AATAAA or a reported variation of this) is situated.

Reviewer #3:

“Taylor et al. reported the observation that the loss of SFPQ in neurons led to an accumulation of intron-containing transcripts in neurite. If true, this result may help understand the role of SFPQ in ALS. However, more analyses are needed to strengthen this conclusion. In addition, while such an association is interesting, in the absence of more mechanistic data, the authors need to tune down many of the interpretation/claims”

Comments:

The authors need to tune down many of the interpretation/claims, including the claim in the abstract that “neurite-centred perturbation of alternatively spliced isoforms initiates the neurodegenerative process.”

We acknowledge these comments also made by reviewer #2 and have modified the manuscript title, abstract and discussion sections accordingly.

One of the key new results is that SFPQ affects local transcriptome in the neurite. However, the analyses of the neurite RNA-seq data can be strengthened, including 1) quality control showing that neurite enrichment actually worked, e.g., known neurite-specific transcripts are highly enriched in the neurite sample relative to the cellular samples. 2) scatter plots/volcano plots to show the global trend that there are more genes downregulated than upregulated. While some analyses have been presented, the text in Fig 1F-G are too small to read. Legends seem to be incorrect. Without more thorough validation and more rigorous analyses, I'm not convinced that SFPQ depletion affects the neurite transcriptome. In addition to comparisons of two genotype in the same compartment, neurite-to-cellular comparisons of the same genotype should also be included.

Our apologies for the small font for the axes of Fig. 1F-G. This was also raised by reviewer #1 and has been rectified. The legends have also been revised. In addition to these analyses, we have added lists of the cell compartment GO terms enriched for transcripts enriched in sibling neurite and null neurite compartments compared to their respective cellular compartments (fig. S4E&H). Both lists indicate the sibling and null neurite compartments are enriched for axons.

Furthermore, we include lists of the most highly expressed genes enriched in sibling cellular and neurite compartments (fig. S4F). As expected, many of the top expressed transcripts in the cellular compartment were nuclear, including SnoRNAs. In the neurite compartment they encode classic neuron projection related proteins such as microtubule constituents and microtubule associated proteins.

Furthermore, in our RT-qPCR experiments validating changes in transcript expression, we were able to verify neurite-specific changes drawn from the RNAseq.

As requested, we have added a volcano plot to show the global trend that there are more genes downregulated than upregulated in null neurites compared with sibling neurites (fig. S4G). This data is also presented in Tables S2 and S3.

The calling of retained introns were not done in a rigorously way. While two tools were used (vast and IRFinder) and score cut-off values were cited, no p-value cut-offs were reported. Also what is the overlap of retained introns found by the two tools?

VAST-TOOLS uses MV.[dPsi]_at_0.95 values to determine significant splicing changes between samples. In sibling and null neurite sample comparisons we set a MV.[dPsi]_at_0.95 cut-off of ≥ 0.01 for events to qualify as changing in null.

In IRFinder, sequencing reads from the three biological replicates for each sample were merged to provide sufficient depth for comparative analysis. IRFinder was performed according to the developer's recommendation and significantly changing events were selected using a p-value cut-off of < 0.05 .

This information is now stated in the relevant results section as well as the relevant supplementary table figure legends.

We find limited overlap between the two tools with 10 common IR events that increase in null neurites. Although there is considerable difference in the introns identified by each tool, the pattern of intron retention is strikingly similar. With both tools, most intron retention changes in null neurites were increases (VAST: 195 inc. vs 35 dec; IRFinder 446 inc. vs 86 dec), and most of these changes were neurite-specific in both cases (Fig. 2B; fig. S7A).

Furthermore, we manually checked mapped reads in IGV for intron-retaining transcripts changing between sibling and null neurites to verify that each tool identified true events.

Line 192: “Intronic polyadenylation (IpA) is an uncommon subclass of ApA reported to occur in immune cells and otherwise has only been associated with cancer” – this is not true. IpA is common outside of immune and cancer cells. Similarly line 296-298. Also intron retention/polyadenylation has been linked to neurodegeneration.

The text here has been modified (line 201 and line 310).

Line 246: tune down: “...uncovering neurite PreT-IR as a player in pathology”. The results showed only indicate an association.

The text here has been modified (line 269).

Line 278-286: I do not see why it is necessary to invoke the controversial theory of local splicing in the neurite. Unlike XBPI whose splicing is independent of the spliceosome, these retained introns contain the canonical splice sites thus are canonical substrates of the nuclear spliceosome.

We agree with the reviewer and this section of discussion has been removed.

The same group reported last year that SFPQ represses cryptic last exons, many of which are found in introns. Please comment on the overlap and discuss possible connections.

CLE transcripts are formed by abnormal splicing of the intronic sequence, leading to the use of a cryptic exon inside the annotated introns. PreT-IR retains the intronic sequence but terminates prematurely at an iPAS. Cryptic last exons (CLEs) form inside introns located anywhere in the gene. PreT-IR are generated in very long introns often located in 5' half of the gene. Both strongly bind SFPQ.

The introns affected are largely distinct. There is a very small number of introns forming CLEs in 24hpf whole *sfpq* embryos that are PreT-IR in our neurite-specific dataset. SFPQ is required to avoid CLEs as well as PreT-IR. Overall, neurons are more prone to PreT-IR than CLEs, probably because they express a much higher number of transcripts from genes containing long introns. The connections between the two phenomena is the importance of the binding of SFPQ to introns for their proper regulation and interpretation by the splicing machinery. We now added a short section on this in our discussion (Lines 345-353).

REVIEWER COMMENTS

Reviewer #1 (Remarks to the Author):

Mutation of SFPQ is associated with motor degeneration and maturation deficits that occur in ALS. As a splice regulator, it has been thought that SFPQ mutation leads to aberrant splicing and accumulation of toxic aggregating products that lead to neurodegeneration. In this work, the authors explore the mechanism by which SFPQ loss of function leads to neurodegeneration in vivo and in cell culture models. In this revised manuscript the authors have thoughtfully addressed concerns raised in the initial review by revising the text to include qualification of conclusions, where needed, and the addition of new supporting data for their model linking iPAS usage to neurodegeneration. These results are exiting and provide a potential mechanism for neurite susceptibility namely, production of truncated mRNAs that are transported/accumulate in neurites where they produce local interfering truncated peptides that cause degeneration. Overall, the manuscript is well written, the data are clearly presented, with appropriate numbers and statistical analyses throughout, and support the main conclusions, which will be of broad interest to those investigating RNA regulation in neurodevelopment and pathological states leading to neurodegeneration, such as ALS. There are a few minor typos and some panels that would benefit from use of a larger font size as there appears to be a conversion issue or simply the font is too small. These minor points are detailed below.

Comments:

- 1) Larger font in panels 1F, 2G, 3B,C, and S4 panels E,H and S5 A and D would be helpful as these are either too small or are not converting well.
- 2) Line 217 "that" is needed before "much larger"
- 3) Line 244 "magnitude" should be "magnitudes" or a specific magnitude should be specified.
- 4) Line 251 "patient" should be "patients"
- 5) Line 315 "truncated" rather than "truncates"
- 6) For the figures with stacks, please indicate the "thickness" of the stack in terms of microns/interval of slices projected in the figure legend.

Reviewer #2 (Remarks to the Author):

The authors have responded to the reviews thoughtfully, and crucially, by adding new data validating some of the key claims of the study.

To me the manuscript is suitable for publication and will make an important contribution to the field.

The title as it is written is a bit of a mouthful and at the very least needs a hyphen for "axon-localised," but perhaps could benefit from a re-think to see if the key message could be conveyed more succinctly, not that I have an immediate suggestion!

Reviewer #3 (Remarks to the Author):

The authors have addressed all of my concerns.

We kindly thank the reviewers for their time in looking at our manuscript, their helpful comments and endorsing its publication in Nature Communications. Below we detail the revisions we have made in response to the remaining points raised.

Reviewer #1

“Mutation of SFPQ is associated with motor degeneration and maturation deficits that occur in ALS. As a splice regulator, it has been thought that SFPQ mutation leads to aberrant splicing and accumulation of toxic aggregating products that lead to neurodegeneration. In this work, the authors explore the mechanism by which SFPQ loss of function leads to neurodegeneration in vivo and in cell culture models.

In this revised manuscript the authors have thoughtfully addressed concerns raised in the initial review by revising the text to include qualification of conclusions, where needed, and the addition of new supporting data for their model linking iPAS usage to neurodegeneration. These results are exiting and provide a potential mechanism for neurite susceptibility namely, production of truncated mRNAs that are transported/accumulate in neurites where they produce local interfering truncated peptides that cause degeneration.

Overall, the manuscript is well written, the data are clearly presented, with appropriate numbers and statistical analyses throughout, and support the main conclusions, which will be of broad interest to those investigating RNA regulation in neurodevelopment and pathological states leading to neurodegeneration, such as ALS.

There are a few minor typos and some panels that would benefit from use of a larger font size as there appears to be a conversion issue or simply the font is too small. These minor points are detailed below.”

Comments:

1) Larger font in panels 1F, 2G, 3B,C, and S4 panels E,H and S5 A and D would be helpful as these are either too small or are not converting well.

The specified panels are now all correctly formatted in line with Nature Communications specifications and are provided at higher resolution to overcome conversion issues. Larger font size has also been used.

2) Line 217 “that” is needed before “much larger”

3) Line 244 “magnitude” should be “magnitudes” or a specific magnitude should be specified.

4) Line 251 “patient” should be “patients”

5) Line 315 “truncated” rather than “truncates”

These sentences have been revised to address the reviewer’s comment. Thank you for spotting.

6) For the figures with stacks, please indicate the “thickness” of the stack in terms of microns/interval of slices projected in the figure legend.

Stack thicknesses have now been included in the relevant figure legends.

Reviewer #2

“The authors have responded to the reviews thoughtfully, and crucially, by adding new data validating some of the key claims of the study.

To me the manuscript is suitable for publication and will make an important contribution to the field.

The title as it is written is a bit of a mouthful and at the very least needs a hyphen for "axon-localised," but perhaps could benefit from a re-think to see if the key message could be conveyed more succinctly, not that I have an immediate suggestion!”

The title has been revised.

Reviewer #3

“The authors have addressed all of my concerns.”